# Dynamic Feature Selection for Efficient and Interpretable Human Activity Recognition

## Abstract

In many machine learning tasks, input features with varying degrees of predictive capability are usually acquired at some cost. For example, in human activity recognition (HAR) and mobile health (mHealth) applications, monitoring performance should be achieved with a low cost to gather different sensory features, as maintaining sensors incur monetary, computation, and energy cost. We propose an adaptive feature selection method that dynamically selects features for prediction at any given time point. We formulate this problem as an $\ell_0$ minimization problem across time, and cast the combinatorial optimization problem into a stochastic optimization formulation. We then utilize a differentiable relaxation to make the problem amenable to gradient-based optimization. Our evaluations on four activity recognition datasets show that our method achieves a favorable trade-off between performance and the number of features used. Moreover, the dynamically selected features of our approach are shown to be interpretable and associated with the actual activity types.

## 1 Introduction

Acquiring predictive features is critical for building trustworthy machine learning systems, but this often comes at a daunting cost. Such a cost can be in the form of energy needed to maintain an ambient sensor (Ardywibowo et al., 2019; Yang et al., 2020), time needed to complete an experiment (Kiefer, 1959), or manpower required to monitor a hospital patient (Pierskalla & Brailer, 1994). Therefore, it becomes important not only to maintain good performance in the specified task, but also a low cost to gather these features.

Indeed, existing Human Activity Recognition (HAR) methods typically use a fixed set of sensors, potentially collecting redundant features to discriminate contexts (Shen & Varshney, 2013; Aziz et al., 2016; Ertuğrul & Kaya, 2017; Cheng et al., 2018). Classic feature selection methods such as the LASSO and its variants can address the performance-cost trade-off by optimizing an objective penalized by a term that helps promote feature sparsity (Tibshirani, 1996; Friedman et al., 2010, 2008; Zou & Hastie, 2005). Such feature selection formulations are often static, that is, a fixed set of features are selected *a priori*. However, different features may offer different predictive power under different contexts. For example, a health worker may not need to monitor a recovering patient as frequently compared to a patient with the declining condition; an experiment performed twice may be redundant; or a smartphone sensor may be predictive when the user is walking but not when the user is in a car. By adaptively selecting which sensor(s) to observe at any given time point, one can further reduce the inherent cost for prediction and achieve a better trade-off between cost and prediction accuracy.

In addition to cost-efficiency, an adaptive feature selection formulation can also lead to more interpretable and trustworthy predictions. Specifically, the predictions made by the model are only based on the selected features, providing a clear relationship between input features and model predictions. Existing efforts on interpreting models are usually based on some post-analyses of the predictions, including the approaches in (1) visualizing higher level representations or reconstructions of inputs based on them (Li et al., 2016; Mahendran & Vedaldi, 2015), (2) evaluating the sensitivity of predictions to local perturbations of inputs or the input gradients (Selvaraju et al., 2017; Ribeiro et al., 2016), and (3) extracting parts of inputs as justifications for predictions (Lei et al., 2016). Another related but orthogonal direction is model compression of training sparse neural networks

with the goal of memory and computational efficiency (Louizos et al., 2017; Tartaglione et al., 2018; Han et al., 2015). All these works require collecting all features first and provide post-hoc feature relevance justifications or network pruning.

Recent efforts on dynamic feature selection adaptively assign features based on immediate statistics (Gordon et al., 2012; Bloom et al., 2013; Ardywibowo et al., 2019; Zappi et al., 2008), ignoring the information a feature may have on future predictions. Others treat feature selection as a Markov Decision Process (MDP) and use Reinforcement Learning (RL) to solve it (He & Eisner, 2012; Karayev et al., 2013; Kolamunna et al., 2016; Spaan & Lima, 2009; Satsangi et al., 2015; Yang et al., 2020). However, solving the RL objective is not straightforward. Besides being sensitive to hyperparameter settings in general, approximations such as state space discretization and greedy approximations of the combinatorial objective were used to make the RL problem tractable.

To this end, we propose a dynamic feature selection method that can be easily integrated into existing deep architectures and trained from end to end, enabling *task-driven dynamic feature selection*. To achieve this, we define a feature selection module that dynamically selects which features to use at any given time point. We then formulate a sequential combinatorial optimization that minimizes the trade-off between the learning task performance and the number of features selected at each time point. To make this problem tractable, we cast this combinatorial optimization problem into a stochastic optimization formulation. We then adopt a differentiable relaxation of the discrete feature selection variables to make it amenable to stochastic gradient descent based optimization. It therefore can be plugged-in and jointly optimized with state-of-the-art neural networks, achieving task-driven feature selection over time. To show our method's ability to adaptively select features while maintaining good performance, we evaluate it on four time-series activity recognition datasets: the UCI Human Activity Recognition (HAR) dataset (Anguita et al., 2013), the OPPORTUNITY dataset (Roggen et al., 2010), the ExtraSensory dataset (Vaizman et al., 2017), as well as the NTU-RGB-D dataset (Shahroudy et al., 2016).

Several ablation studies and comparisons with other dynamic and static feature selection methods demonstrate the efficacy of our proposed method. Specifically, our dynamic feature selection is able to use as low as 0.28% of the sensor features while still maintaining good human activity monitoring accuracy. Moreover, our dynamically selected features are shown to be interpretable with direct correspondence with different contexts and activity types.

## 2 METHODOLOGY

### 2.1 THE $\ell_0$-NORM MINIMIZATION PROBLEM

Many regularization methods have been developed to solve simultaneous feature selection and model parameter estimation (Tibshirani, 1996; Zou & Hastie, 2005; Tibshirani, 1997; Sun et al., 2014; Simon et al., 2011). The ideal penalty for the purpose of feature selection is the $\ell_0$-norm of the model coefficients for all predictors. This norm is equivalent to the number of nonzero terms in all the model coefficients. Given a dataset $\mathcal{D}$ containing $N$ independent and identically distributed (*iid*) input-output pairs $\{(\boldsymbol{x}_1, \boldsymbol{y}_1), \ldots, (\boldsymbol{x}_N, \boldsymbol{y}_N)\}$ with each $\boldsymbol{x}_i$ containing $P$ features, a hypothesis class of predictor functions $f(\cdot; \boldsymbol{\theta})$, and a loss function $\mathcal{L}(\hat{\boldsymbol{y}}, \boldsymbol{y})$ between prediction $\hat{\boldsymbol{y}}$ and true output $\boldsymbol{y}$, the $\ell_0$-norm regularized optimization problem can be written as follows:

$$\min_{\boldsymbol{\theta}} \quad \frac{1}{N} \left( \sum_{i=1}^{N} \mathcal{L}(f(\boldsymbol{x}_i; \boldsymbol{\theta}), \boldsymbol{y}_i) \right) + \lambda \|\boldsymbol{\theta}\|_0, \tag{1}$$

where $\|\boldsymbol{\theta}\|_0 = \sum_{j=1}^{P} \mathbb{I}[\theta_j \neq 0]$ penalizes the number of nonzero model coefficients.

In the models that linearly transform the input features $\boldsymbol{x}_i$, penalizing the weights relating to each feature in $\boldsymbol{x}_i$ enables sparse feature subset selection. However, such a selection is static, as it does not adaptively select features that are appropriate for a given context. Moreover, the optimization above is computationally prohibitive as it involves combinatorial optimization to select the subset of nonzero model coefficients corresponding to the input features.

In the following, we formulate our adaptive dynamic feature selection problem when learning with multivariate time series. Coupled with training recurrent neural networks, this adaptive feature

selection problem is transformed into a sequential context-dependent feature subset selection problem, to which we devise a stochastic relaxation to make the problem tractable.

## 2.2 Dynamic Feature Selection via Sequential Context-dependent Feature Subset Selection

Instead of finding a subset of nonzero model coefficients, an equivalent formulation can be derived by directly selecting the feature subset. Without loss of generality, let $\boldsymbol{z}$ be a binary vector that indicates whether each feature is selected or not. Then, the original $\ell_0$-norm optimization formulation can be equivalently written as follows:

$$\min_{\boldsymbol{\theta},\boldsymbol{z}} \quad \frac{1}{N}\bigg(\sum_{i=1}^{N}\mathcal{L}(f(\boldsymbol{x}_i\circ\boldsymbol{z};\boldsymbol{\theta}),\boldsymbol{y}_i)\bigg) + \lambda\|\boldsymbol{z}\|_0. \tag{2}$$

Compared to the original problem, the penalty on the number of selected features is through the $\ell_0$-norm of $\boldsymbol{z}$. This formulation is more flexible, as $\boldsymbol{z}$ can be made dependent on corresponding input features, output labels, or any contextual information, allowing us to formulate our dynamic feature selection problem when learning with multivariate time series data. Specifically, let the input-output pairs $(\boldsymbol{x}_i,\boldsymbol{y}_i)$ be a pair of time series data of length $T_i$. At each time $t$, our model predicts the output $\boldsymbol{y}_i^t$, as well as the next feature set to select $\boldsymbol{z}_i^t$. This optimization problem can be formulated as:

$$\min_{\boldsymbol{\theta},\boldsymbol{z}} \quad \frac{1}{N}\bigg(\sum_{i=1}^{N}\sum_{t=1}^{T_i}\mathcal{L}(f(\boldsymbol{x}_i^{0:t-1}\circ\boldsymbol{z}_i^{0:t-1};\boldsymbol{\theta}),\boldsymbol{y}_i^t)\bigg) + \lambda\sum_{i=1}^{N}\sum_{t=1}^{T_i}\|\boldsymbol{z}_i^t\|_0. \tag{3}$$

Here, we are tasked to find a set of parameters $\boldsymbol{\theta}$ and feature sets $\boldsymbol{z}_i^t$ for each sample $i$ at each time point $t$ to optimize the trade-off between model performance and the number of selected features. The model then uses the parameters and the previously observed features $\mathcal{X}_i^t \triangleq \boldsymbol{x}_i^{0:t-1}\circ\boldsymbol{z}_i^{0:t-1}$ to infer the next output $\boldsymbol{y}_i^t$. However, the above formulation remains intractable, as it involves combinatorial optimization to select the feature subsets at each time point, in addition to the joint optimization of the model parameters and variable selection. Naively, one may also need to solve a separate optimization problem to find $\boldsymbol{z}_i^t$ for each time point during the run time. In the following section, we derive a relaxation based on stochastic optimization parameterizing $\boldsymbol{z}_i^t$'s to make the above problem tractable.

## 2.3 Relaxation through Stochastic Optimization

Instead of finding the exact feature subsets indexed by $\boldsymbol{z}_i^t$ that achieve the optimal regularized objective, one can treat these $\boldsymbol{z}_i^t$'s as binary random variables and seek to optimize the distribution $\boldsymbol{\pi}(\boldsymbol{z}|\boldsymbol{\phi})$ that generates these random variables. For the ease of exposition, we first focus on the relaxation of the non-adaptive formulation in (1) as follows:

$$\min_{\boldsymbol{\theta},\boldsymbol{\phi}} \quad \mathbb{E}_{(\boldsymbol{x}_i,\boldsymbol{y}_i)\sim\mathcal{D}}\bigg[\mathbb{E}_{\boldsymbol{z}\sim\boldsymbol{\pi}(\boldsymbol{z}|\boldsymbol{\phi})}\Big[\mathcal{L}(f(\boldsymbol{x}_i\circ\boldsymbol{z};\boldsymbol{\theta}),\boldsymbol{y}_i) + \lambda\|\boldsymbol{z}\|_0\Big]\bigg]. \tag{4}$$

Note that the solution to this problem is equivalent to the original one, as the original combinatorial problem can be recovered by setting $\boldsymbol{\pi}(\boldsymbol{z}|\boldsymbol{\phi}) = \mathrm{Bern}(\boldsymbol{\phi})$, a Bernoulli distribution parameterized by $\boldsymbol{\phi}$, and restricting $\boldsymbol{\phi}\in\{0,1\}$. Using this relaxation, the regularization term can now be evaluated analytically:

$$\mathbb{E}_{\boldsymbol{z}\sim\boldsymbol{\pi}(\boldsymbol{z}|\boldsymbol{\phi})}\Big[\|\boldsymbol{z}\|_0\Big] = \mathbb{E}_{\boldsymbol{z}\sim\mathrm{Bern}(\boldsymbol{\phi})}\Big[\|\boldsymbol{z}\|_0\Big] = \sum_{j=1}^{P}\boldsymbol{\pi}(\boldsymbol{z}|\boldsymbol{\phi})_j = \sum_{j=1}^{P}\phi_j, \tag{5}$$

On the other hand, the outer expectation in (4) can be approximated using minibatches. Relaxation of binary random variables has been adopted in Louizos et al. (2017) for network architecture sparsification, and in Yamada et al. (2019); Balın et al. (2019) for static feature selection. Here, we extend the above relaxation for time series data, where unlike previous works, the binary random variables are parameterized locally and are context-dependent, and features are selected adaptively across time. We first note that our adaptive feature selection formulation in (3) allows each time point to have its own feature selection distribution $\boldsymbol{\pi}_i^t(\boldsymbol{z}|\boldsymbol{\phi}) \triangleq \boldsymbol{\pi}(\boldsymbol{z}|\mathcal{X}_i^{t-1},\boldsymbol{\phi})$ conditioned on previously selected observed features $\mathcal{X}_i^{t-1}$ as defined above. Let $\boldsymbol{\pi}_i(\boldsymbol{z}|\boldsymbol{\phi})$ be the set of $\boldsymbol{\pi}_i^t(\boldsymbol{z}|\boldsymbol{\phi})$ for all $t\in\{1,\dots,T_i\}$. The stochastic relaxation of the adaptive feature selection formulation can be written as follows:

$$\min_{\boldsymbol{\theta},\boldsymbol{\phi}} \quad \mathbb{E}_{(\boldsymbol{x}_i,\boldsymbol{y}_i)\sim\mathcal{D}}\bigg[\mathbb{E}_{\boldsymbol{z}_i\sim\boldsymbol{\pi}_i(\boldsymbol{z}|\boldsymbol{\phi})}\Big[\sum_{t=1}^{T_i}\mathcal{L}(f(\mathcal{X}_i^{t-1};\boldsymbol{\theta}),\boldsymbol{y}_i^t)\Big] + \lambda\sum_{t=1}^{T_i}\sum_{j=1}^{P}\boldsymbol{\pi}_i^t(\boldsymbol{z}|\boldsymbol{\phi})_j\bigg]. \tag{6}$$

## 2.4 MODEL PARAMETERIZATION AND DIFFERENTIABLE RELAXATION

The difficulty in solving the above problem using gradient descent is that the discrete random variables $z_i^t$'s are not directly amenable to stochastic reparameterization techniques. An effective and simple to implement formulation that we adopt is the Gumbel-Softmax reparameterization (Jang et al., 2016; Maddison et al., 2016), which relaxes a discrete valued random variable $z$ parameterized by $\phi$ to a continuous random variable $\tilde{z}$. Firstly, we can parameterize $\boldsymbol{\pi}(z|\mathcal{X}_i^{t-1}, \boldsymbol{\phi})$ using a vector-valued function $\boldsymbol{\sigma}(\mathcal{X}_i^{t-1}, \boldsymbol{\phi})$ of the previous observations $\mathcal{X}_i^{t-1}$, with $\boldsymbol{\phi}$ now being the parameters of $\boldsymbol{\sigma}(\cdot)$. The distribution can now be rewritten as $\boldsymbol{\pi}(z|\mathcal{X}_i^{t-1}, \boldsymbol{\phi}) = \text{Bern}(\boldsymbol{\sigma}(\mathcal{X}_i^{t-1}, \boldsymbol{\phi}))$. With this, the discrete valued random variables $z_i^t$ can be relaxed into continuous random variables $\tilde{z}_i^t$ as follows:

$$\tilde{z}_i^t = \frac{1}{1 + \exp\left(-(\log \boldsymbol{\sigma}(\mathcal{X}_i^{t-1}, \boldsymbol{\phi}) + L)/\tau\right)}. \tag{7}$$

Here, $L = \log u - \log(1-u)$ is a logistic distribution, where $u \sim \text{Unif}(0, 1)$, and $\tau$ is a temperature parameter. For low values of $\tau$, $\tilde{z}_i^t$ approaches a sample of a binary random variable, recovering the original discrete problem, while for high values, $\tilde{z}_i^t$ will equal $\frac{1}{2}$. With this, we are able to compute gradient estimates of $\tilde{z}_i^t$ and approximate the gradient of $z_i^t$ as $\nabla_{\boldsymbol{\theta}, \boldsymbol{\phi}} z_i^t \approx \nabla_{\boldsymbol{\theta}, \boldsymbol{\phi}} \tilde{z}_i^t$. This enables us to backpropagate through the discrete random variables and train the selection parameters along with the model parameters jointly using stochastic gradient descent. Meanwhile, at test time, we sample binary random variables from the learned probabilities.

## 2.5 MODEL SPECIFICATION

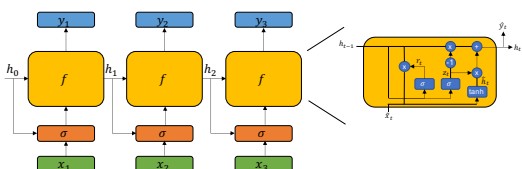

To complete our formulation, we specify the model architecture that we use. We have implemented our adaptive dynamic feature selection with a Gated Recurrent Unit (GRU) (Cho et al., 2014a), a type of Recurrent Neural Network (RNN) (Graves et al., 2013), as shown in Figure 1. Here, we have the previous observations $\mathcal{X}_i^{t-1}$ being summarized by the hidden state $\boldsymbol{h}_i^{t-1}$. For

Figure 1: The proposed Gated Recurrent Unit (GRU) based architecture for our adaptive monitoring model specification. Here, features at any given time are selected based on the previous observations summarized by $h_{t-1}$.

adaptive feature selection, the selection distribution is made dependent on $\boldsymbol{h}_i^{t-1}$ using a sigmoid of its linear transformation by a weight matrix $W$ as follows: $\boldsymbol{\sigma}(\mathcal{X}_i^{t-1}, \boldsymbol{\phi}) = \text{SIGMOID}(W\boldsymbol{h}_i^{t-1})$, such that $\boldsymbol{\phi} = \{W\}$. We note that such a module can be easily integrated into many existing deep architectures and trained from end to end, allowing for *task-driven feature selection*. For example, the module can be applied to Recurrent Convolutional Neural Networks (RCNN) (Liang & Hu, 2015) to selectively determine which convolutional patches/channels to use, or to general feedforward networks to selectively deactivate certain neurons/channels to reduce computation. We have demonstrated this ability by applying it to an Independent RNN (Li et al., 2018) benchmarked on the NTU-RGB-D dataset (Shahroudy et al., 2016), as detailed in Appendix A.4.

With the model specified, our method can be applied to existing human activity recognition datasets. Specifically, we are now able to train a prediction model and dynamic feature selection policy offline, and test it on a withheld testing set. The application of our model to online learning is subject to future work.

## 3 RELATED WORK

Existing HAR systems typically use a fixed set of sensors, potentially collecting redundant features for easily discriminated contexts. Methods that attempt to find a fixed or static feature set often rank feature sets using metrics such as Information Gain (Shen & Varshney, 2013), or relevancy ranking through a filtering strategy (Aziz et al., 2016; Ertuğrul & Kaya, 2017; Cheng et al., 2018). However, static feature selection can potentially result in collecting redundant information for highly discriminable contexts.

Work on dynamic feature selection can be divided into Reinforcement Learning (RL) based and non-RL approaches. Non-RL based approaches vary from assigning certain features to certain activities

(Gordon et al., 2012), pre-defining feature subsets for prediction (Bloom et al., 2013; Strubell et al., 2015), optimizing the trade-off between prediction entropy and the number of selected features (Ardywibowo et al., 2019), to building a metaclassifier for sensor selection (Zappi et al., 2008). These methods all use immediate rewards to perform feature selection. For predicting long activity sequences, this potentially ignores the information that a feature may have on future predictions, or conversely, overestimate the importance of a feature given previous observations.

Among the RL based approaches, some methods attempt to build an MDP to decide which feature to select next or whether to stop acquiring features and make a prediction (He & Eisner, 2012; Karayev et al., 2013; Kolamunna et al., 2016). These methods condition the choice of one feature on the observation generated by another one, instead of choosing between all sensors simultaneously. Spaan & Lima (2009) and Satsangi et al. (2015) formulated a Partially Observable MDP (POMDP) using a discretization of the continuous state to model the policy. Yang et al. (2020) formulate an RL objective by penalizing the prediction performance by the number of sensors used. Although using a desirable objective, the method employs a greedy maximization process to approximately solve the combinatorial optimization. Moreover, they do not integrate easily with existing deep architectures.

Attention is another method worth noting, as it is able to select the most relevant segments of a sequence for the current prediction (Vaswani et al., 2017). Attention modules have been recently used for activity recognition (Ma et al., 2019). However, like most attention methods, it requires all of the features to be observed before deciding which features are the most important for prediction. Moreover, the number of instances attended to is not penalized. Finally, soft attention methods typically weight the inputs, instead of selecting the feature subset. Indeed, our experiments on naively applying attention for dynamic feature selection show that it always selects 100% of the features at all times.

Sparse regularization has previously been formulated for deep models, e.g., Liu et al. (2015); Louizos et al. (2017); Frankle & Carbin (2018), but their focus has primarily been in statically compressing model sizes or reducing overfitting, instead of dynamically selecting features for prediction. In particular, $\ell_1$ regularization is a common method to promote feature sparsity (Tibshirani, 1996; Friedman et al., 2010, 2008; Zou & Hastie, 2005).

Selection or skipping along the temporal direction to decide when to memorize vs update model state has been considered in Hu et al. (2019); Campos et al. (2018); Neil et al. (2016). These works either are not context dependent or do not consider energy efficiency or interpretability. Additionally, skipping time steps may not be suitable for continuous monitoring tasks including HAR, where we are tasked to give a prediction at every time step. Nevertheless, our dynamic/adaptive feature selection is orthogonal to temporal selection/skipping and we leave exploring the potential integration of these two directions as our future research.

Finally, there have been many formulations that propose to solve the issue of backpropagation through discrete random variables (Jang et al., 2016; Maddison et al., 2016; Tucker et al., 2017; Grathwohl et al., 2017; Yin & Zhou, 2018). REBAR (Tucker et al., 2017) and RELAX (Grathwohl et al., 2017) employ REINFORCE and introduce relaxation-based baselines to reduce sample variance of the estimator. However, these baseline functions increase the computation and cause potential conflict between minimizing the sample variance of the gradient estimate and maximizing the expectation objective. Augment-REINFORCE-Merge is a self-control gradient estimator that does not need additional baselines (Yin & Zhou, 2018). It provides unbiased gradient estimates that exhibit low variance, but its direct application to autoregressive or sequential setups is not addressed by Yin & Zhou (2018) and leads to approximate gradients. Moreover, an exact sequential formulation will require prohibitive computation, squared in sequence length forward passes.

## 4 EXPERIMENTS

**Benchmark Datasets and Performance Evaluation** We evaluate our model on four different datasets: the UCI Human Activity Recognition (HAR) using Smartphones Dataset (Anguita et al., 2013), the OPPORTUNITY Dataset (Roggen et al., 2010), the ExtraSensory dataset (Vaizman et al., 2017), and the NTU-RGB-D dataset (Shahroudy et al., 2016). Although there are many other human activity recognition benchmark datasets (Chen et al., 2020), we choose the above datasets to better convey our message of achieving feature usage efficiency and interpretability using our adaptive

feature selection framework with the following reasons. First, the UCI HAR dataset is a clean dataset with no missing values, allowing us to benchmark different methods without any discrepancies in data preprocessing confounding our evaluations. Second, the OPPORTUNITY dataset contains activity labels that correspond to specific sensors. An optimal adaptive feature selector should primarily choose these sensors under specific contexts with clear physical meaning. Finally, the ExtraSensory dataset studies a multilabel classification problem, where two or more labels can be active at any given time, while the NTU-RGB-D dataset is a complicated activity recognition dataset with over 60 classes of activities using data from 25 skeleton joints. These datasets allow us to benchmark model performance in a complex setting. For all datasets, we randomly split data both chronologically and by different subjects. More details for each dataset and its corresponding experiment setup is provided under its own subheading in the following and also in Appendix A. Due to the page limit, our implementation details and results on the NTU-RGB-D dataset are available in Appendix A and B.

We investigate several aspects of our model performance on these benchmarks. To show the effect in prediction accuracy when our selection module is considered, we compare its performance to a standard GRU network (Cho et al., 2014b). To show the effect of considering dynamic feature selection, we compare a nonadaptive $\ell_0$ formulation that statically selects features by solving (4) (Louizos et al., 2017). The performance of our $\ell_0$ regularized formulation is also benchmarked with an $\ell_1$ regularized formulation. To benchmark the performance of our differentiable relaxation-based optimization strategy, we implement the Straight-Through estimator (Hinton et al., 2012) and Augment-REINFORCE-Merge (ARM) gradient estimates (Yin & Zhou, 2018) as alternative methods to optimize our formulation. As stated in the previous section, the fully sequential application of ARM was not addressed in the original paper, and will be prohibitively expensive to compute exactly. Hence, we combine ARM and Straight-Through (ST) estimator (Hinton et al., 2012) as another approach to optimize our formulation. More specifically, we calculate the gradients with respect to the Bernoulli variables with ARM, and use the ST estimator to backpropagate the gradients through the Bernoulli variables to previous layers' parameters. We also have tested different values for the temperature hyperparameter $\tau$ in Appendix D, where we observe that the settings with the temperature parameters below 1 generally yield the best results with no noticeable performance difference.

To further show the importance of considering the sparse regularized formulation, we compare with an attention-based feature selection, selecting features based on the largest attention weights. Because attention yields feature attention weights instead of feature subsets, we select features by using a hard threshold $\alpha$ of the attention weights and scaling the selected features by $1 - \alpha$ for different values of $\alpha$. Indeed, without this modification, we observe that an attention-based feature selection would select 100% of the features at all times.

Finally, we have attempted to implement the dynamic feature selection method by Yang et al. (2020) as a distinctly different benchmark. However, without any implementation details provided by the authors, we were not able to reproduce their results.

**UCI HAR Dataset** We first test our proposed method on performing simultaneous prediction and adaptive feature selection on the UCI HAR dataset (Anguita et al., 2013). This dataset consists of 561 smartphone sensor measurements including various gyroscope and accelerometer readings, with the task of inferring the activity that the user performs at any given time. There are six possible activities

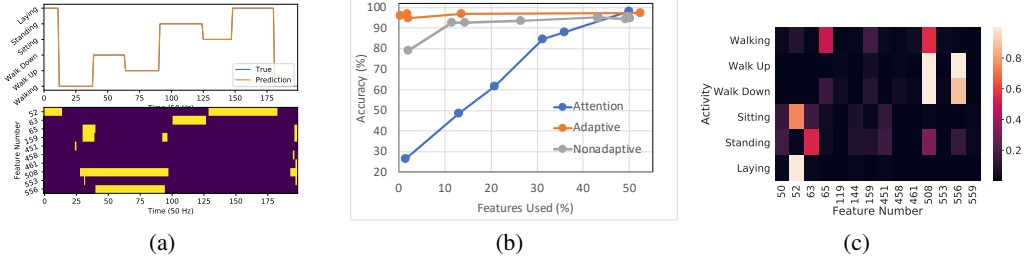

(a)           (b)           (c)

Figure 2: UCI HAR Dataset results: (a) Prediction and features selected of the proposed model $\lambda = 1$. (b) Feature selection vs. accuracy trade-off curve comparison. (c) Heatmap of sensor feature activations under each activity of the UCI HAR dataset. Only active features are shown out of the 561 features in total.

Table 1: Comparison of various optimization techniques for our model on the UCI HAR dataset.
*Accuracies and average number of features selected are in (%).

| Method | Accuracy | Feat. Selected |
|---|---|---|
| Gumbel-Softmax $\lambda = 1$ | **97.18** | **0.28** |
| ARM $\lambda = 1$ (Yin & Zhou, 2018) | 95.73 | 11.67 |
| ST-ARM $\lambda = 1$ (Yin & Zhou, 2018; Hinton et al., 2012) | 92.79 | 1.92 |
| Straight Through $\lambda = 1$ (Hinton et al., 2012) | 89.38 | 0.31 |
| L1 Regularization $\lambda = 1$ | 90.43 | 19.48 |

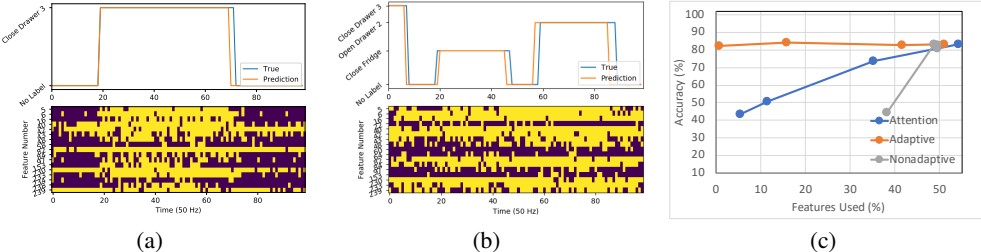

|  (a)  |  (b)  |  (c)  |

Figure 3: OPPORTUNITY Dataset results: (a) Prediction and features selected of the proposed model $\lambda = 1$. (b) Prediction and features selected of the proposed model on a set of activity transitions. (c) Feature selection vs. Error trade-off curve comparison.

that a subject can perform: walking, walking upstairs, walking downstairs, sitting, standing, and laying.

We first compare various optimization methods, using stochastic gradients by differential relaxation using Gumbel-Softmax reparametrization, ARM, ST-ARM, Straight-Through gradients, and an $\ell_1$ regularized formulation to solve adaptive feature selection. The results are provided in Table 1. As shown, Gumbel-Softmax achieves the best prediction accuracy with the least number of features. Utilizing either the Straight Through estimator, ARM, or ST-ARM for gradient estimation cannot provide a better balance between accuracy and efficiency compared with the Gumbel-Softmax relaxation-based optimization. Indeed, the performance of the ST estimator is expected, as there is a mismatch between the forward propagated activations and the backward propagated gradients in the estimator. Meanwhile, we attribute the lower performance of the ARM and ST-ARM optimizer to its use in a sequential fashion, which was not originally considered. The lower performance of the $\ell_1$ regularized formulation is expected, as $\ell_1$ regularization is an approximation to the problem of selecting the optimal feature subset. In the following experiments, we have seen similar trends and only report the results from the Gumbel-Softmax based optimization.

Benchmarking results of different models are given in Table 2. As shown, our adaptive feature selection model is able to achieve a competitive accuracy using only 0.28% of the features, or on average about 1.57 sensors at any given time. We also observe that both the attention and our adaptive formulation is able to improve upon the accuracy of the standard GRU, suggesting that feature selection can also regularize the model to improve accuracy. Although the attention-based model yields the best accuracy, this comes at a cost of utilizing around 50% of the features at any given time. We also have checked the average accuracy of our model on a time-aligned testing set to show that our model is stable for long-term predictions in Appendix E.

We study the effect of the regularization weight $\lambda$ by varying it from $\lambda \in \{1, 0.1, 0.01, 0.005, 0.001\}$. We compare this with the attention model by varying the threshold $\alpha$ used to select features from $\alpha \in \{0.5, 0.9, 0.95, 0.99, 0.995, 0.999\}$, as well as the nonadaptive model by varying its $\lambda$ from $\lambda \in \{1000, 100, \dots 0.01, 0.005, 0.001\}$. A trade-off curve between the number of selected features and the performance for the three models can be seen in Figure 2(b). As shown in the figure, the accuracy of the attention model suffers increasingly with smaller feature subsets, as attention is not a formulation specifically tailored to find sparse solutions. On the other hand, the accuracy of our adaptive formulation is unaffected by the number of features, suggesting that selecting around 0.3% of the features on average may be optimal for the given problem. It further confirms that our adaptive formulation selects the most informative features given the context. The performance of

Table 2: Comparison of various models for adaptive monitoring on three activity recognition datasets.
*Accuracy metrics and average number of features selected are all in (%).

| Method | UCI HAR | | OPPORTUNITY | | ExtraSensory | | |
|---|---|---|---|---|---|---|---|
| | Accuracy | Features | Accuracy | Features | Accuracy | F1 | Features |
| Adaptive (Ours) $\lambda = 1$ | 97.18 | **0.28** | **84.26** | **15.88** | **91.14** | **55.06** | **11.25** |
| Attention $\alpha = 0.5$ | **98.38** | 49.94 | 83.42 | 54.20 | 90.37 | 53.29 | 54.73 |
| Nonadaptive $\lambda = 1$ (Louizos et al., 2017) | 95.49 | 14.35 | 81.63 | 49.57 | 91.13 | 53.18 | 42.32 |
| No selection (GRU) (Cho et al., 2014b) | 96.67 | 100 | 84.16 | 100 | 91.14 | 53.53 | 100 |

the nonadaptive model is consistent for feature subsets of size 10% or greater. However, it suffers a drop in accuracy for extremely small feature subsets. This shows that for static selection, selecting a feature set that is too large would result in collecting many redundant features for certain contexts, while selecting a feature set that is too small would be insufficient for maintaining accuracy.

An example of dynamically selected features can be seen in Figure 2(a). We plot the prediction of our model compared to the true label and illustrate the features that are used for prediction. We also plot a heatmap for the features selected under each activity in Figure 2(c). Although these features alone may not be exclusively attributed as the only features necessary for prediction under specific activities, such a visualization is useful to retrospectively observe the features selected by our model at each time-point. Note that mainly 5 out of the 561 features are used for prediction at any given time. Observing the selected features, we see that for the static activities such as sitting, standing, and laying, only sensor feature 52 and 63, features relating to the gravity accelerometer, are necessary for prediction. On the other hand, the active states such as walking, walking up, and walking down requires 3 sensor features: sensor 65, 508, and 556, which are related to both the gravity accelerometer and the body accelerometer. This is intuitively appealing as, under the static contexts, the body accelerometer measurements would be relatively constant, and unnecessary for prediction. On the other hand, for the active contexts, the body accelerometer measurements are necessary to reason about how the subject is moving and accurately discriminate between the different active states. Meanwhile, we found that measurements relating to the gyroscope were unnecessary for prediction.

**UCI OPPORTUNITY Dataset**   We further test our proposed method on the UCI OPPORTUNITY Dataset (Roggen et al., 2010). This dataset consists of multiple different label types for human activity, ranging from locomotion, hand gestures, to object interactions. The dataset consists of 242 measurements from accelerometers and Inertial Measurement Units (IMUs) attached to the user, as well as accelerometers attached to different objects with which the user can interact.

We use the mid-level gesture activities as the target for our models to predict, which contain gestures related to specific objects, such as opening a door and drinking from a cup. A comparison of the accuracy and the percentage of selected features by different models is given in Table 2, while example predictions and a trade-off curve are constructed and shown in Figures 3(a), 3(b), and 3(c), with a similar trend as the results on the UCI HAR dataset. Notably, the trade-off for the nonadaptive models remains constant for $\lambda \in \{0.0001, 0.001, \ldots, 1\}$, with a sharp decrease in accuracy for $\lambda \geq 10$.

A heatmap for the selected features under each activity is shown in Figure 4. Here, the active sensor features across all activities are features 40 and 42, readings of the IMU attached to the subject's back, feature 82, readings from the IMU attached to the left upper arm (LUA), and features 230 and 239, location tags that estimate the subject's position. We posit that these general sensor features are selected to track the subject's overall position and movements, as they are also predominantly selected in cases with no labels. Meanwhile, sensors 5, 6, and 16, readings from the accelerometer attached to the hip, LUA, and back, are specific to activities involving opening/closing doors or drawers.

Interestingly, sensors attached to specific objects, such as accelerometers on doors and cups, are unnecessary for prediction. We attribute this to the severe amount of missing values of these sensors. Indeed, the sensors that have the least amount of missing values are the body sensors and the localization tags. We hypothesize that the model prefers these sensors for their consistent discriminative power on multiple activity types compared to the object specific sensors. In addition to these object specific sensors, 5 IMUs, 9 accelerometers, and 2 localization tags can be completely turned off without significantly affecting prediction performance on this task.

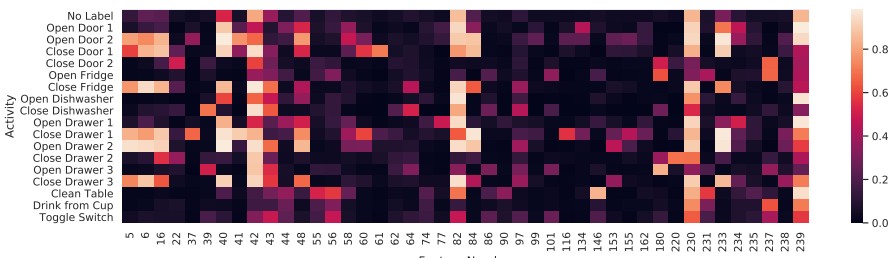

Figure 4: Heatmap of sensor feature activations under each activity of the OPPORTUNITY dataset. *Only active features are shown out of the 242 features in total.

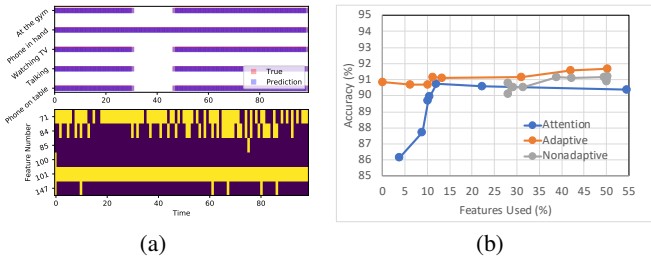

(a)                 (b)

Figure 5: ExtraSensory Dataset results: (a) Prediction and features selected of the proposed model. (b) Feature selection vs. Error trade-off curve comparison.

**ExtraSensory Dataset** We further test our proposed method on the ExtraSensory Dataset (Vaizman et al., 2017). This is a multilabel classification dataset, where two or more labels can be active at any given time. It consists of 51 different context labels, and 225 sensor features. We frame the problem as a multilabel binary classification problem, where we have a binary output for each label indicating whether it is active. A comparison of the accuracy and selected features by different models tested can be seen in Table 2. Our method is again competitive with the standard GRU model using less than 12% of all the features.

A trade-off curve is shown in Figure 5(b), where we see a similar trend for both adaptive and attention models. However we were unable to obtain a feature selection percentage lower than 25% for the nonadaptive model even with $\lambda$ as large as $10^4$. We believe that this is because at least 25% of statically selected features are needed; otherwise the nonadaptive model will degrade in performance catastrophically, similar to the OPPORTUNITY dataset results. A heatmap and detailed discussion of the features that our model dynamically selected can be found in Appendix C.

The results on these three datasets along with the results on the NTU-RGB-D dataset in Appendix B indicate that our adaptive monitoring framework provides the best trade-off between feature efficiency and accuracy, while the features that it dynamically selects are also interpretable and associated with the actual activity types.

## 5 CONCLUSIONS

We propose a novel method for performing adaptive feature selection by *sequential context-dependent feature subset selection*, which is cast into a stochastic optimization formulation by modifying the $\ell_0$ regularized minimization formulation. To make this problem tractable, we perform a stochastic relaxation along with a differentiable reparamaterization, making the optimization amenable to gradient-based optimization with auto-differentiation. We apply this method to human activity recognition by implementing our method to Recurrent Neural Network-based architectures. We benchmark our model on four different activity recognition datasets and have compared it with various adaptive and static feature selection benchmarks. Our results show that our model maintains a desirable prediction performance using a fraction of the sensors or features. The features that our model selected were shown to be interpretable and associated with the activity types.

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

## A    IMPLEMENTATION DETAILS

Here, we provide the implementation details for the reported results in each benchmark dataset. In general, the added computation and memory incurred by our adaptive monitoring framework are insignificant, as it consists of only an additional fully connected layer used to infer the next feature set. This would only add extra $H \times P$ parameters and multiply-add operations, where $H$ is the number of hidden neurons and $P$ is the number of input features. This additional computational burden is insignificant compared to the memory and computational cost of the main network, which are typically of order higher than $O(HP)$.

### A.1    UCI HAR DATASET

The UCI HAR dataset consists of a training set and a testing set. To implement our adaptive feature selection and other baseline methods, we divide the training set into a separate validation set consisting of 2 subjects. We preprocess the data by normalizing it with the mean and standard deviation. We then divide the instances of each subject into segments of length 200.

The base model we utilize is a one-layer GRU with 2800 neurons for the hidden state. We use the cross-entropy of the predicted vs. actual labels as the performance measure. We use a temperature of 0.05 for the Gumbel-Softmax relaxation. We optimize this with a batch size of 10 using the RMSProp optimizer, setting the learning rate to $10^{-4}$ and the smoothing constant to 0.99 for 3000 epochs. We then save both the latest model and the best model validated on the validation set.

### A.2    OPPORTUNITY DATASET

The OPPORTUNITY dataset consists of multiple demonstrations of different activity types. We first extract the instances into segments containing no missing labels for the mid-level gestures. Segments of length smaller than 100 are padded using the observed values at the next time-points in the instance. We then normalize the data such that its values are between -1 and 1. The authors of the dataset recommended removing some features that they believed are not useful, however we find that this does not affect performance and instead use the entire feature set. We have also experimented with interpolating the missing values but also find that it does not affect performance compared to imputing the missing values with zeros. Using this, we randomly shuffle the segments and assign 80% for training, 10% for validation, and 10% for testing.

The base model we utilize is a two-layer GRU with 256 neurons for each layer's hidden state. The cross-entropy of the predicted vs. actual labels is adopted as the performance measure. We use a temperature of 0.05 for the Gumbel-Softmax relaxation. We do not include the cross-entropy loss for the time points with missing labels. We also scale the total performance loss of the observed labels for each batch by $\frac{\#\text{timepoints}}{\#\text{labelled timepoints}}$. We optimize this loss with a batch size of 100 using the RMSProp optimizer, setting the learning rate to $10^{-4}$ and the smoothing constant to 0.99 for 3000 epochs. We then save both the latest model and the best model validated on the validation set.

### A.3    EXTRASENSORY DATASET

The ExtraSensory dataset consists of multiple demonstrations of human behavior under different activities, where two or more activity labels can be active at the same time. We first extract the instances into segments containing no missing labels for the middle level gestures. Segments of length smaller than 70 are padded using the observed values at the next time-points in the instance. We then normalize the data such that its values are in between -1 and 1. We have experimented with interpolating the missing values but also find that it does not affect performance compared to imputing the missing values with zeros. Using this, we randomly shuffle the segments and assign 70% for training, 10% for validation, and 20% for testing.

The base model we utilize is a one-layer GRU with 2240 neurons for its hidden state. We use a temperature of 0.05 for the Gumbel-Softmax relaxation. We use the binary cross-entropy of the predicted vs. actual labels as the performance measure, where the model outputs a binary decision for each label, representing whether each label is active or not. We do not include the performance loss for the missing labels and scale the total performance loss of the observed labels for each batch

by $\frac{\#\text{timepoints}\times\#\text{total labels}}{\#\text{observed labels in labelled timepoints}}$. We optimize this scaled loss with a batch size of 100 using the RMSProp optimizer, setting the learning rate to $10^{-4}$ and the smoothing constant to 0.99 for 10000 epochs. We then save both the latest model and the best model validated on the validation set.

## A.4 NTU-RGB-D DATASET

We first preprocess the NTU-RGB-D dataset to remove all the samples with missing skeleton data. We then segment the time-series skeleton data across subjects into 66.5% training, 3.5% validation, and 30% testing sets. The baseline model that we have implemented for the NTU-RGB-D dataset is the Independent RNN (Li et al., 2018). This model consists of stacked RNN modules with several additional dropout, batch normalization, and fully connected layers in between. Our architecture closely follows the densely connected independent RNN of Li et al. (2018). To incorporate feature selection using either our adaptive formulation or an attention-based formulation, we add an additional RNN to the beginning of this model. This RNN takes as input the 25 different joint features and is tasked to select the joints to use for prediction further along the architecture pipeline. Since the joints are in the form of 3D coordinates, our feature selection method is modified such that it selects either all 3 of the X, Y, and Z coordinates of a particular joint, or none at all. Our architecture can be seen in Figure 6.

Similar as the baseline method presented in Li et al. (2018), we have trained this architecture using a batch size of 128 and a sequence length of 20 using the Adam optimizer with a patience threshold of 100 iterations. We then save both the latest model and the best model validated on the validation set.

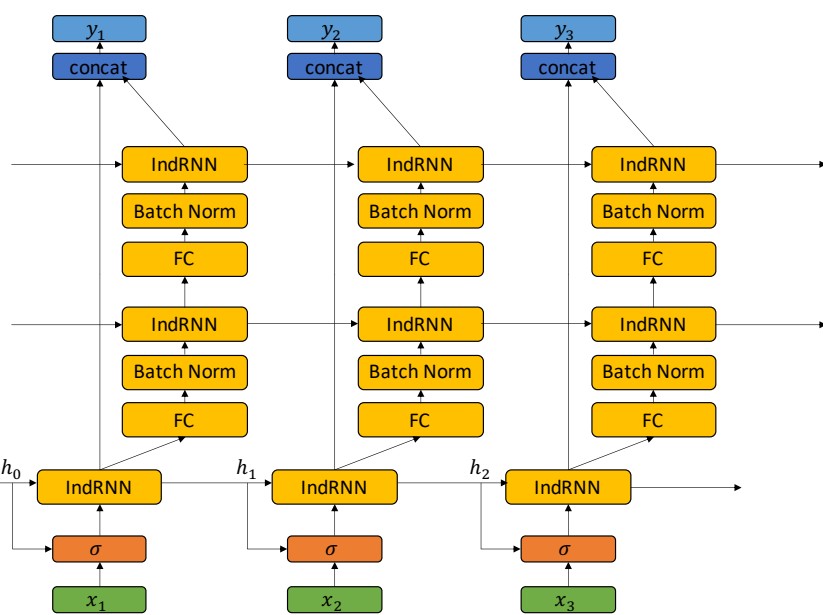

Figure 6: Our modified densely connected independent RNN architecture for adaptive feature selection.

## B RESULTS AND DISCUSSION OF THE NTU-RGB-D DATASET

We have tested our proposed method on the NTU-RGB-D dataset (Shahroudy et al., 2016). This dataset consists of 60 different activities performed by either a single individual or two individuals. The measurements of this dataset are in the form of skeleton data consisting of 25 different 3D coordinates of the corresponding joints of the participating individuals.

We compare our method with three different baselines shown in Table 3: the standard independent RNN, a soft attention baseline, and a thresholded attention baseline. We see that our method maintains a competitive accuracy compared to the baseline using less than 50% of the features. On the other hand, because the thresholded attention formulation is not specifically optimized for feature sparsity,

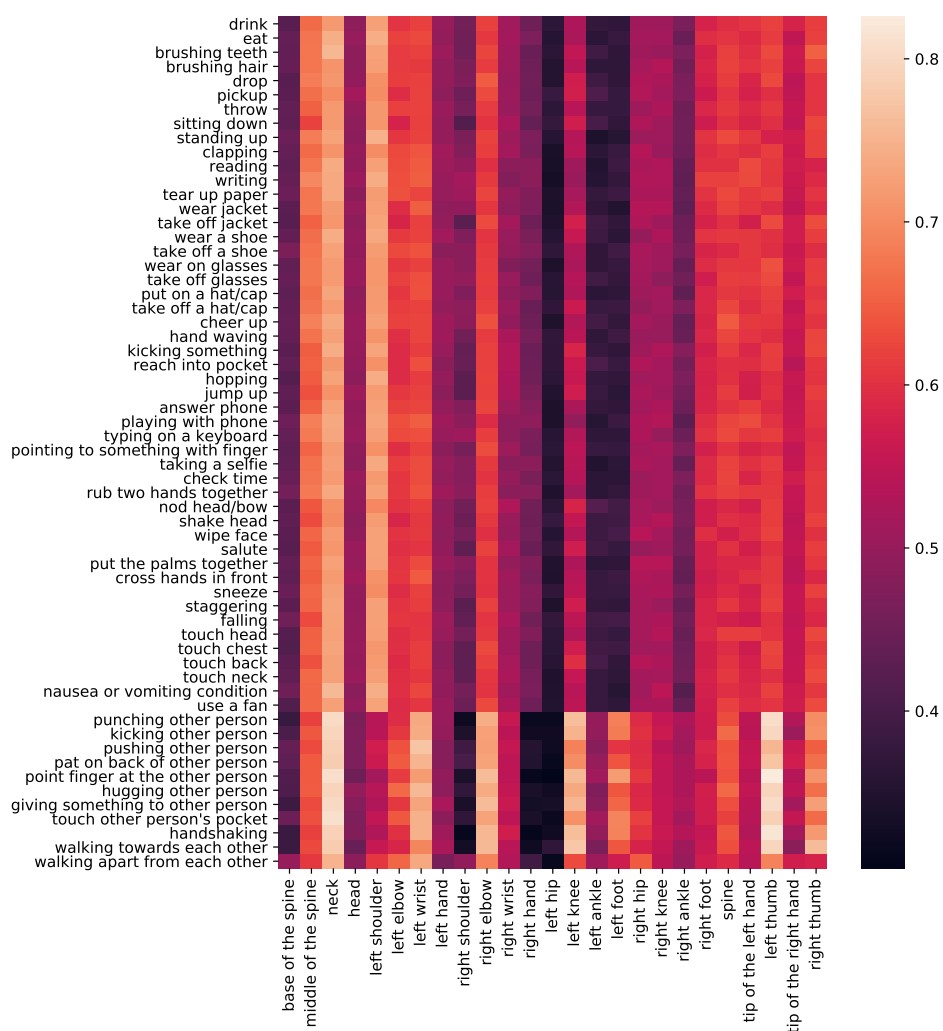

Figure 7: Heatmap of sensor feature activations under each activity state of the NTU-RGB-D dataset.

we see that it performs significantly worse compared to the other methods. Meanwhile, the soft-attention slightly improves upon the accuracy of the base architecture. However, as also indicated by our other experiments, soft-attention is not a dynamic feature selection method, and tends to select 100% of the features at all times.

A heatmap for the features selected under each activity is shown in Figure 7. Here, we can see that there are two distinct feature sets used for two different types of interactions: single person interactions and two person interactions. Indeed, since the two person activities require sensor measurements from two individuals, the dynamic feature selection would need to prioritize different features to observe their activities as opposed to single person activities.

Table 3: Comparison of various methods for activity recognition on the NTU-RGB-D dataset. *Accuracies and average number of features selected are in (%).

| Method | Accuracy | Features Selected |
|---|---|---|
| Adaptive | 80.54 | **49.65** |
| Thresholded attention | 40.07 | 52.31 |
| Soft attention | **83.28** | 100 |
| No selection | 83.02 | 100 |

## C  RESULTS AND DISCUSSION OF THE EXTRASENSORY DATASET

Figure 8: Heatmap of sensor feature activations under each activity state of the ExtraSensory dataset.

A heatmap of the features selected under each activity state can be seen in Figure 8. As shown, there are four groups of sensor features that are used across activities: the phone magnetometer (57-71), watch accelerometer magnitude (85-88), watch accelerometer direction (101-105), and location (138-147). For two particular states, 'on a bus' and 'drinking alcohol', phone accelerometer measurements (5-52) become necessary for prediction. Some states such as 'at home', 'at main workplace', and 'phone in pocket' are notably sparse in sensor feature usage. We believe that these states are static, and do not require much sensor usage to monitor effectively. Other sensors such as the phone gyroscope, phone state, audio measurements and properties, compass, and various low-frequency sensors are largely unnecessary for prediction in this dataset.

## D  EFFECTS OF THE HYPERPARAMETER $\tau$ ON MODEL PERFORMANCE

We observe the effects of the temperature hyperparameter in (7) on our model's performance. To do this, we have tested several hyperparameter values in our experiment with the UCI HAR dataset. The results of our tests can be seen in Figure 9. In general, the settings with the temperature parameters below 1 generally yield the best results with no noticeable performance difference. Once the temperature is set to above 1, we observe a sharp increase in errors. We attribute this to the mismatch between training and testing setups, where in testing, discrete binary values are sampled while in training, the samples are reduced to an equal weighting between the features.

## E  MODEL PERFORMANCE AND STABILITY ACROSS TIME

We show the average accuracy over every 1000 seconds of running the model on the testing subjects in the UCI HAR dataset in Table 4. Based on the performance of the model across time, the model is shown to be stable for long-term predictions. In general, there is no clear temporal degradation in the testing performance for this dataset. Instead, the change of prediction errors is mostly dependent on the underlying activity types.

Table 4: The average model performance across time averaged across time-aligned testing subjects.

| Time | 0-999 | 1000-1999 | 2000-2999 | 3000-3999 |
|---|---|---|---|---|
| Error (%) | 3.49 | 2.93 | 6.46 | 4.06 |
| Std. Dev. | 1.89 | 1.23 | 1.05 | 1.67 |

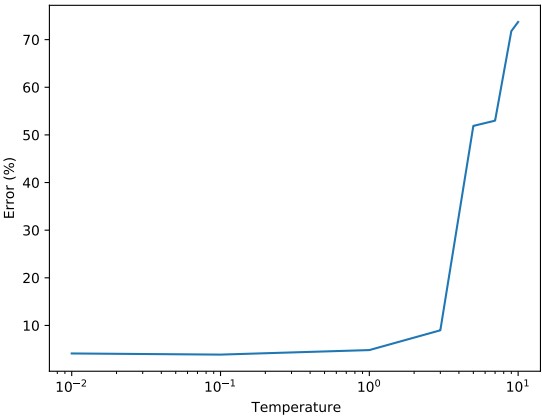

Figure 9: The effect of the temperature hyperparameter $\tau$ on the performance of the model.

## F   UNION OF ALL FEATURES SELECTED BY THE ADAPTIVE MODEL

Here, in addition to showing the average number of selected features, we compute the percentage of all features considered by our model across the full time-length. In other words, the results presented here show the union of selected features across the time horizon. In Section 4, we chose to present the average number of selected features as it directly reflects the number of required sensors for accurate HAR. Hence, it clearly shows the benefits of our proposed dynamic/adaptive feature selection with respect to the power usage for sensor data collection. From Table 5, it is clear that the percentage of all the features considered across the full time-length is also significantly low for each of the three benchmark datasets, which further validates the potential of our dynamic feature selection even when additional operational cost of turning on/off sensors needs to be considered.

Table 5: The percentage of the union of selected features across three benchmark datasets.

| Dataset | (%) Union |
|---|---|
| UCI HAR | 3.56 |
| OPPORTUNITY | 19.83 |
| ExtraSensory | 26.66 |

# DYNAMIC FEATURE SELECTION FOR EFFICIENT AND INTERPRETABLE HUMAN ACTIVITY RECOGNITION

**Anonymous authors**

## ABSTRACT

In many machine learning tasks, input features with varying degrees of predictive capability are usually acquired at some cost. For example, in human activity recognition (HAR) and mobile health (mHealth) applications, monitoring performance should be achieved with a low cost to gather different sensory features, as maintaining sensors incur monetary, computation, and energy cost. We propose an adaptive feature selection method that dynamically selects features for prediction at any given time point. We formulate this problem as an $\ell_0$ minimization problem across time, and cast the combinatorial optimization problem into a stochastic optimization formulation. We then utilize a differentiable relaxation to make the problem amenable to gradient-based optimization. Our evaluations on four activity recognition datasets show that our method achieves a favorable trade-off between performance and the number of features used. Moreover, the dynamically selected features of our approach are shown to be interpretable and associated with the actual activity types.

## 1 INTRODUCTION

Acquiring predictive features is critical for building trustworthy machine learning systems, but this often comes at a daunting cost. Such a cost can be in the form of energy needed to maintain an ambient sensor [1, 2], time needed to complete an experiment [3], or manpower required to monitor a hospital patient [4]. Therefore, it becomes important not only to maintain good performance in the specified task, but also a low cost to gather these features.

Indeed, existing Human Activity Recognition (HAR) methods typically use a fixed set of sensors, potentially collecting redundant features to discriminate contexts [5, 6, 7, 8]. Classic feature selection methods such as the LASSO and its variants can address the performance-cost trade-off by optimizing an objective penalized by a term that helps promote feature sparsity [9, 10, 11, 12]. Such feature selection formulations are often static, that is, a fixed set of features are selected *a priori*. However, different features may offer different predictive power under different contexts. For example, a health worker may not need to monitor a recovering patient as frequently compared to a patient with the declining condition; an experiment performed twice may be redundant; or a smartphone sensor may be predictive when the user is walking but not when the user is in a car. By adaptively selecting which sensor(s) to observe at any given time point, one can further reduce the inherent cost for prediction and achieve a better trade-off between cost and prediction accuracy.

In addition to cost-efficiency, an adaptive feature selection formulation can also lead to more interpretable and trustworthy predictions. Specifically, the predictions made by the model are only based on the selected features, providing a clear relationship between input features and model predictions. Existing efforts on interpreting models are usually based on some post-analyses of the predictions, including the approaches in (1) visualizing higher level representations or reconstructions of inputs based on them [13, 14], (2) evaluating the sensitivity of predictions to local perturbations of inputs or the input gradients [15, 16], and (3) extracting parts of inputs as justifications for predictions [17]. Another related but orthogonal direction is model compression of training sparse neural networks with the goal of memory and computational efficiency [18, 19, 20]. All these works require collecting all features first and provide post-hoc feature relevance justifications or network pruning.

Recent efforts on dynamic feature selection adaptively assign features based on immediate statistics [21, 22, 1, 23], ignoring the information a feature may have on future predictions. Others treat

feature selection as a Markov Decision Process (MDP) and use Reinforcement Learning (RL) to solve it [24, 25, 26, 27, 28, 2]. However, solving the RL objective is not straightforward. Besides being sensitive to hyperparameter settings in general, approximations such as state space discretization and greedy approximations of the combinatorial objective were used to make the RL problem tractable.

To this end, we propose a dynamic feature selection method that can be easily integrated into existing deep architectures and trained from end to end, enabling *task-driven dynamic feature selection*. To achieve this, we define a feature selection module that dynamically selects which features to use at any given time point. We then formulate a sequential combinatorial optimization that minimizes the trade-off between the learning task performance and the number of features selected at each time point. To make this problem tractable, we cast this combinatorial optimization problem into a stochastic optimization formulation. We then adopt a differentiable relaxation of the discrete feature selection variables to make it amenable to stochastic gradient descent based optimization. It therefore can be plugged-in and jointly optimized with state-of-the-art neural networks, achieving task-driven feature selection over time. To show our method's ability to adaptively select features while maintaining good performance, we evaluate it on four time-series activity recognition datasets: the UCI Human Activity Recognition (HAR) dataset [29], the OPPORTUNITY dataset [30], the ExtraSensory dataset [31], as well as the NTU-RGB-D dataset [32].

Several ablation studies and comparisons with other dynamic and static feature selection methods demonstrate the efficacy of our proposed method. Specifically, our dynamic feature selection is able to use as low as 0.28% of the sensor features while still maintaining good human activity monitoring accuracy. Moreover, our dynamically selected features are shown to be interpretable with direct correspondence with different contexts and activity types.

## 2 METHODOLOGY

### 2.1 THE $\ell_0$-NORM MINIMIZATION PROBLEM

Many regularization methods have been developed to solve simultaneous feature selection and model parameter estimation [9, 12, 33, 34, 35]. The ideal penalty for the purpose of feature selection is the $\ell_0$-norm of the model coefficients for all predictors. This norm is equivalent to the number of nonzero terms in all the model coefficients. Given a dataset $\mathcal{D}$ containing $N$ independent and identically distributed (*iid*) input-output pairs $\{(\boldsymbol{x}_1, \boldsymbol{y}_1), \ldots, (\boldsymbol{x}_N, \boldsymbol{y}_N)\}$ with each $\boldsymbol{x}_i$ containing $P$ features, a hypothesis class of predictor functions $f(\cdot; \boldsymbol{\theta})$, and a loss function $\mathcal{L}(\hat{\boldsymbol{y}}, \boldsymbol{y})$ between prediction $\hat{\boldsymbol{y}}$ and true output $\boldsymbol{y}$, the $\ell_0$-norm regularized optimization problem can be written as follows:

$$\min_{\boldsymbol{\theta}} \quad \frac{1}{N}\left(\sum_{i=1}^{N} \mathcal{L}(f(\boldsymbol{x}_i; \boldsymbol{\theta}), \boldsymbol{y}_i)\right) + \lambda\|\boldsymbol{\theta}\|_0, \tag{1}$$

where $\|\boldsymbol{\theta}\|_0 = \sum_{j=1}^{P} \mathbb{I}[\theta_j \neq 0]$ penalizes the number of nonzero model coefficients.

In the models that linearly transform the input features $\boldsymbol{x}_i$, penalizing the weights relating to each feature in $\boldsymbol{x}_i$ enables sparse feature subset selection. However, such a selection is static, as it does not adaptively select features that are appropriate for a given context. Moreover, the optimization above is computationally prohibitive as it involves combinatorial optimization to select the subset of nonzero model coefficients corresponding to the input features.

In the following, we formulate our adaptive dynamic feature selection problem when learning with multivariate time series. Coupled with training recurrent neural networks, this adaptive feature selection problem is transformed into a sequential context-dependent feature subset selection problem, to which we devise a stochastic relaxation to make the problem tractable.

### 2.2 DYNAMIC FEATURE SELECTION VIA SEQUENTIAL CONTEXT-DEPENDENT FEATURE SUBSET SELECTION

Instead of finding a subset of nonzero model coefficients, an equivalent formulation can be derived by directly selecting the feature subset. Without loss of generality, let $\boldsymbol{z}$ be a binary vector that indicates whether each feature is selected or not. Then, the original $\ell_0$-norm optimization formulation can be equivalently written as follows:

$$\min_{\boldsymbol{\theta}, \boldsymbol{z}} \quad \frac{1}{N} \left( \sum_{i=1}^{N} \mathcal{L}(f(\boldsymbol{x}_i \circ \boldsymbol{z}; \boldsymbol{\theta}), \boldsymbol{y}_i) \right) + \lambda \|\boldsymbol{z}\|_0. \tag{2}$$

Compared to the original problem, the penalty on the number of selected features is through the $\ell_0$-norm of $\boldsymbol{z}$. This formulation is more flexible, as $\boldsymbol{z}$ can be made dependent on corresponding input features, output labels, or any contextual information, allowing us to formulate our dynamic feature selection problem when learning with multivariate time series data. Specifically, let the input-output pairs $(\boldsymbol{x}_i, \boldsymbol{y}_i)$ be a pair of time series data of length $T_i$. At each time $t$, our model predicts the output $\boldsymbol{y}_i^t$, as well as the next feature set to select $\boldsymbol{z}_i^t$. This optimization problem can be formulated as:

$$\min_{\boldsymbol{\theta}, \boldsymbol{z}} \quad \frac{1}{N} \left( \sum_{i=1}^{N} \sum_{t=1}^{T_i} \mathcal{L}(f(\boldsymbol{x}_i^{0:t-1} \circ \boldsymbol{z}_i^{0:t-1}; \boldsymbol{\theta}), \boldsymbol{y}_i^t) \right) + \lambda \sum_{i=1}^{N} \sum_{t=1}^{T_i} \|\boldsymbol{z}_i^t\|_0. \tag{3}$$

Here, we are tasked to find a set of parameters $\boldsymbol{\theta}$ and feature sets $\boldsymbol{z}_i^t$ for each sample $i$ at each time point $t$ to optimize the trade-off between model performance and the number of selected features. The model then uses the parameters and the previously observed features $\mathcal{X}_i^t \triangleq \boldsymbol{x}_i^{0:t-1} \circ \boldsymbol{z}_i^{0:t-1}$ to infer the next output $\boldsymbol{y}_i^t$. However, the above formulation remains intractable, as it involves combinatorial optimization to select the feature subsets at each time point, in addition to the joint optimization of the model parameters and variable selection. Naively, one may also need to solve a separate optimization problem to find $\boldsymbol{z}_i^t$ for each time point during the run time. In the following section, we derive a relaxation based on stochastic optimization parameterizing $\boldsymbol{z}_i^t$'s to make the above problem tractable.

### 2.3 Relaxation through Stochastic Optimization

Instead of finding the exact feature subsets indexed by $\boldsymbol{z}_i^t$ that achieve the optimal regularized objective, one can treat these $\boldsymbol{z}_i^t$'s as binary random variables and seek to optimize the distribution $\boldsymbol{\pi}(\boldsymbol{z}|\boldsymbol{\phi})$ that generates these random variables. For the ease of exposition, we first focus on the relaxation of the non-adaptive formulation in (1) as follows:

$$\min_{\boldsymbol{\theta}, \boldsymbol{\phi}} \quad \mathbb{E}_{(\boldsymbol{x}_i, \boldsymbol{y}_i) \sim \mathcal{D}} \left[ \mathbb{E}_{\boldsymbol{z} \sim \boldsymbol{\pi}(\boldsymbol{z}|\boldsymbol{\phi})} \left[ \mathcal{L}(f(\boldsymbol{x}_i \circ \boldsymbol{z}; \boldsymbol{\theta}), \boldsymbol{y}_i) + \lambda \|\boldsymbol{z}\|_0 \right] \right]. \tag{4}$$

Note that the solution to this problem is equivalent to the original one, as the original combinatorial problem can be recovered by setting $\boldsymbol{\pi}(\boldsymbol{z}|\boldsymbol{\phi}) = \text{Bern}(\boldsymbol{\phi})$, a Bernoulli distribution parameterized by $\boldsymbol{\phi}$, and restricting $\boldsymbol{\phi} \in \{0, 1\}$. Using this relaxation, the regularization term can now be evaluated analytically:

$$\mathbb{E}_{\boldsymbol{z} \sim \boldsymbol{\pi}(\boldsymbol{z}|\boldsymbol{\phi})} \left[ \|\boldsymbol{z}\|_0 \right] = \mathbb{E}_{\boldsymbol{z} \sim \text{Bern}(\boldsymbol{\phi})} \left[ \|\boldsymbol{z}\|_0 \right] = \sum_{j=1}^{P} \boldsymbol{\pi}(\boldsymbol{z}|\boldsymbol{\phi})_j = \sum_{j=1}^{P} \phi_j, \tag{5}$$

On the other hand, the outer expectation in (4) can be approximated using minibatches. To extend the above relaxation for time series data, we first note that our adaptive feature selection formulation in (3) allows each time point to have its own feature selection distribution $\boldsymbol{\pi}_i^t(\boldsymbol{z}|\boldsymbol{\phi}) \triangleq \boldsymbol{\pi}(\boldsymbol{z}|\mathcal{X}_i^{t-1}, \boldsymbol{\phi})$ conditioned on previous observations $\mathcal{X}_i^{t-1}$. Let $\boldsymbol{\pi}_i(\boldsymbol{z}|\boldsymbol{\phi})$ be the set of $\boldsymbol{\pi}_i^t(\boldsymbol{z}|\boldsymbol{\phi})$ for all $t \in \{1, \dots, T_i\}$. The stochastic relaxation of the adaptive feature selection formulation can be written as follows:

$$\min_{\boldsymbol{\theta}, \boldsymbol{\phi}} \quad \mathbb{E}_{(\boldsymbol{x}_i, \boldsymbol{y}_i) \sim \mathcal{D}} \left[ \mathbb{E}_{\boldsymbol{z}_i \sim \boldsymbol{\pi}_i(\boldsymbol{z}|\boldsymbol{\phi})} \left[ \sum_{t=1}^{T_i} \mathcal{L}(f(\mathcal{X}_i^{t-1}; \boldsymbol{\theta}), \boldsymbol{y}_i^t) \right] + \lambda \sum_{t=1}^{T_i} \sum_{j=1}^{P} \boldsymbol{\pi}_i^t(\boldsymbol{z}|\boldsymbol{\phi})_j \right]. \tag{6}$$

### 2.4 Model Parameterization and Differentiable Relaxation

The difficulty in solving the above problem using gradient descent is that the discrete random variables $\boldsymbol{z}_i^t$'s are not directly amenable to stochastic reparameterization techniques. An effective and simple to implement formulation that we adopt is the Gumbel-Softmax reparameterization [36, 37], which relaxes a discrete valued random variable $z$ parameterized by $\phi$ to a continuous random variable $\tilde{z}$. Firstly, we can parameterize $\boldsymbol{\pi}(\boldsymbol{z}|\mathcal{X}_i^{t-1}, \boldsymbol{\phi})$ using a vector-valued function $\boldsymbol{\sigma}(\mathcal{X}_i^{t-1}, \boldsymbol{\phi})$ of the previous observations $\mathcal{X}_i^{t-1}$, with $\boldsymbol{\phi}$ now being the parameters of $\boldsymbol{\sigma}(\cdot)$. The distribution can now be rewritten as $\boldsymbol{\pi}(\boldsymbol{z}|\mathcal{X}_i^{t-1}, \boldsymbol{\phi}) = \text{Bern}(\boldsymbol{\sigma}(\mathcal{X}_i^{t-1}, \boldsymbol{\phi}))$. With this, the discrete valued random variables $\boldsymbol{z}_i^t$ can be relaxed into continuous random variables $\tilde{\boldsymbol{z}}_i^t$ as follows:

$$\tilde{\boldsymbol{z}}_i^t = \frac{1}{1 + \exp\left(-(\log \boldsymbol{\sigma}(\mathcal{X}_i^{t-1}, \boldsymbol{\phi}) + L)/\tau\right)}. \tag{7}$$

Here, $L = \log u - \log(1 - u)$ is a logistic distribution, where $u \sim \text{Unif}(0, 1)$, and $\tau$ is a temperature parameter. For low values of $\tau$, $\tilde{z}_i^t$ approaches a sample of a binary random variable, recovering the original discrete problem, while for high values, $\tilde{z}_i^t$ will equal $\frac{1}{2}$. With this, we are able to compute gradient estimates of $\tilde{z}_i^t$ and approximate the gradient of $z_i^t$ as $\nabla_{\boldsymbol{\theta}, \boldsymbol{\phi}} z_i^t \approx \nabla_{\boldsymbol{\theta}, \boldsymbol{\phi}} \tilde{z}_i^t$. This enables us to backpropagate through the discrete random variables and train the selection parameters along with the model parameters jointly using stochas-

tic gradient descent. Meanwhile, at test time, we sample binary random variables from the learned probabilities.

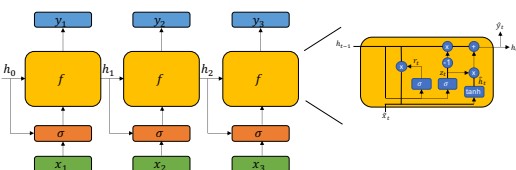

## 2.5 MODEL SPECIFICATION

To complete our formulation, we specify the model architecture that we use. We have implemented our adaptive dynamic feature selection with a Gated Recurrent Unit (GRU) [38], a type of Recurrent Neural Network (RNN) [39], as shown in Figure 1. Here, we have the previous

Figure 1: The proposed Gated Recurrent Unit (GRU) based architecture for our adaptive monitoring model specification. Here, features at any given time are selected based on the previous observations summarized by $h_{t-1}$.

observations $\mathcal{X}_i^{t-1}$ being summarized by the hidden state $\boldsymbol{h}_i^{t-1}$. For adaptive feature selection, the selection distribution is made dependent on $\boldsymbol{h}_i^{t-1}$ using a sigmoid of its linear transformation by a weight matrix $W$ as follows: $\boldsymbol{\sigma}(\mathcal{X}_i^{t-1}, \boldsymbol{\phi}) = \text{SIGMOID}(W\boldsymbol{h}_i^{t-1})$, such that $\boldsymbol{\phi} = \{W\}$. We note that such a module can be easily integrated into many existing deep architectures and trained from end to end, allowing for *task-driven feature selection*. For example, the module can be applied to Recurrent Convolutional Neural Networks (RCNN) [40] to selectively determine which convolutional patches/channels to use, or to general feedforward networks to selectively deactivate certain neurons/channels to reduce computation. We have demonstrated this ability by applying it to an Independent RNN [41] benchmarked on the NTU-RGB-D dataset [32], as detailed in Appendix A.4.

## 3 RELATED WORK

Existing HAR systems typically use a fixed set of sensors, potentially collecting redundant features for easily discriminated contexts. Methods that attempt to find a fixed or static feature set often rank feature sets using metrics such as Information Gain [5], or relevancy ranking through a filtering strategy [6, 7, 8]. However, static feature selection can potentially result in collecting redundant information for highly discriminable contexts.

Work on dynamic feature selection can be divided into Reinforcement Learning (RL) based and non-RL approaches. Non-RL based approaches vary from assigning certain features to certain activities [21], pre-defining feature subsets for prediction [22, 42], optimizing the trade-off between prediction entropy and the number of selected features [1], to building a metaclassifier for sensor selection [23]. These methods all use immediate rewards to perform feature selection. For predicting long activity sequences, this potentially ignores the information that a feature may have on future predictions, or conversely, overestimate the importance of a feature given previous observations.

Among the RL based approaches, some methods attempt to build an MDP to decide which feature to select next or whether to stop acquiring features and make a prediction [24, 25, 26]. These methods condition the choice of one feature on the observation generated by another one, instead of choosing between all sensors simultaneously. Spaan and Lima [27] and Satsangi et al. [28] formulated a Partially Observable MDP (POMDP) using a discretization of the continuous state to model the policy. Yang et al. [2] formulate an RL objective by penalizing the prediction performance by the number of sensors used. Although using a desirable objective, the method employs a greedy maximization process to approximately solve the combinatorial optimization. Moreover, they do not integrate easily with existing deep architectures.

Attention is another method worth noting, as it is able to select the most relevant segments of a sequence for the current prediction [43]. Attention modules have been recently used for activity recognition [44]. However, like most attention methods, it requires all of the features to be observed before deciding which features are the most important for prediction. Moreover, the number of instances attended to is not penalized. Finally, soft attention methods typically weight the inputs,

instead of selecting the feature subset. Indeed, our experiments on naively applying attention for dynamic feature selection show that it always selects 100% of the features at all times.

Sparse regularization has previously been formulated for deep models, e.g., [45, 18, 46], but their focus has primarily been in statically compressing model sizes or reducing overfitting, instead of dynamically selecting features for prediction. In particular, $\ell_1$ regularization is a common method to promote feature sparsity [9, 10, 11, 12].

Finally, there have been many formulations that propose to solve the issue of backpropagation through discrete random variables [36, 37, 47, 48, 49]. REBAR [47] and RELAX [48] employ REINFORCE and introduce relaxation-based baselines to reduce sample variance of the estimator. However, these baseline functions increase the computation and cause potential conflict between minimizing the sample variance of the gradient estimate and maximizing the expectation objective. Augment-REINFORCE-Merge is a self-control gradient estimator that does not need additional baselines [49]. It provides unbiased gradient estimates that exhibit low variance, but its direct application to autoregressive or sequential setups is not addressed by Yin and Zhou [49] and leads to approximate gradients. Moreover, an exact sequential formulation will require prohibitive computation, squared in sequence length forward passes.

## 4 EXPERIMENTS

**Benchmark Datasets and Performance Evaluation**   We evaluate our model on four different datasets: the UCI Human Activity Recognition (HAR) using Smartphones Dataset [29], the OPPOR-TUNITY Dataset [30], the ExtraSensory dataset [31], and the NTU-RGB-D dataset [32]. Although there are many other human activity recognition benchmark datasets [50], we choose the above datasets to better convey our message of achieving feature usage efficiency and interpretability using our adaptive feature selection framework with the following reasons. First, the UCI HAR dataset is a clean dataset with no missing values, allowing us to benchmark different methods without any discrepancies in data preprocessing confounding our evaluations. Second, the OPPORTUNITY dataset contains activity labels that correspond to specific sensors. An optimal adaptive feature selector should primarily choose these sensors under specific contexts with clear physical meaning. Finally, the ExtraSensory dataset studies a multilabel classification problem, where two or more labels can be active at any given time, while the NTU-RGB-D dataset is a complicated activity recognition dataset with over 60 classes of activities using data from 25 skeleton joints. These datasets allow us to benchmark model performance in a complex setting. Due to the page limit, our implementation details and results on the NTU-RGB-D dataset are available in Appendix A and B.

We investigate several aspects of our model performance on these benchmarks. To show the effect in prediction accuracy when our selection module is considered, we compare its performance to a standard GRU network [51]. To show the effect of considering dynamic feature selection, we compare a nonadaptive $\ell_0$ formulation that statically selects features by solving (4) [18]. The performance of our $\ell_0$ regularized formulation is also benchmarked with an $\ell_1$ regularized formulation. To benchmark the performance of our differentiable relaxation-based optimization strategy, we implement the Straight-Through estimator [52] and Augment-REINFORCE-Merge (ARM) gradient estimates [49] as alternative methods to optimize our formulation. As stated in the previous section, the fully sequential application of ARM was not addressed in the original paper, and will be prohibitively

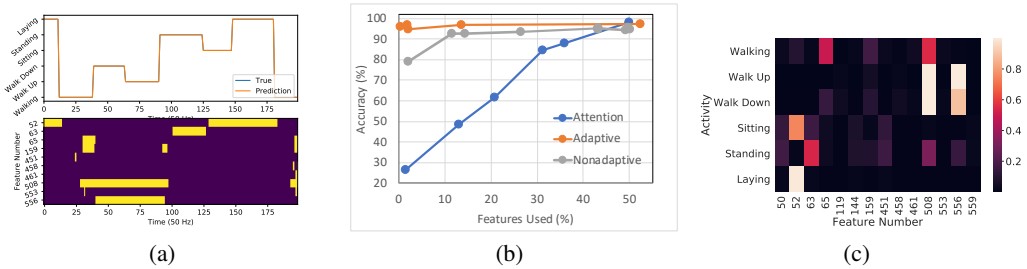

|     (a)     |     (b)     |     (c)     |

Figure 2: UCI HAR Dataset results: (a) Prediction and features selected of the proposed model $\lambda = 1$. (b) Feature selection vs. accuracy trade-off curve comparison. (c) Heatmap of sensor feature activations under each activity of the UCI HAR dataset. Only active features are shown out of the 561 features in total.

expensive to compute exactly. Hence, we combine ARM and Straight-Through (ST) estimator [52] as another approach to optimize our formulation. More specifically, we calculate the gradients with respect to the Bernoulli variables with ARM, and use the ST estimator to backpropagate the gradients through the Bernoulli variables to previous layers' parameters.

To further show the importance of considering the sparse regularized formulation, we compare with an attention-based feature selection, selecting features based on the largest attention weights. Because attention yields feature attention weights instead of feature subsets, we select features by using a hard threshold $\alpha$ of the attention weights and scaling the selected features by $1 - \alpha$ for different values of $\alpha$. Indeed, without this modification, we observe that an attention-based feature selection would select 100% of the features at all times.

Finally, we have attempted to implement the dynamic feature selection method by Yang et al. [2] as a distinctly different benchmark. However, without any implementation details provided by the authors, we were not able to reproduce their results.

**UCI HAR Dataset** We first test our proposed method on performing simultaneous prediction and adaptive feature selection on the UCI HAR dataset [29]. This dataset consists of 561 smartphone sensor measurements including various gyroscope and accelerometer readings, with the task of inferring the activity that the user performs at any given time. There are six possible activities that a subject can perform: walking, walking upstairs, walking downstairs, sitting, standing, and laying.

Table 1: Comparison of various optimization techniques for our model on the UCI HAR dataset. *Accuracies and average number of features selected are in (%).

| Method | Accuracy | Feat. Selected |
|---|---|---|
| Gumbel-Softmax $\lambda = 1$ | **97.18** | **0.28** |
| ARM $\lambda = 1$ [49] | 95.73 | 11.67 |
| ST-ARM $\lambda = 1$ [49, 52] | 92.79 | 1.92 |
| Straight Through $\lambda = 1$ [52] | 89.38 | 0.31 |
| L1 Regularization $\lambda = 1$ | 90.43 | 19.48 |

We first compare various optimization methods, using stochastic gradients by differential relaxation using Gumbel-Softmax reparametrization, ARM, ST-ARM, Straight-Through gradients, and an $\ell_1$ regularized formulation to solve adaptive feature selection. The results are provided in Table 1. As shown, Gumbel-Softmax achieves the best prediction accuracy with the least number of features. Utilizing either the Straight Through estimator, ARM, or ST-ARM for gradient estimation cannot provide a better balance between accuracy and efficiency compared with the Gumbel-Softmax relaxation-based optimization. Indeed, the performance of the ST estimator is expected, as there is a mismatch between the forward propagated activations and the backward propagated gradients in the estimator. Meanwhile, we attribute the lower performance of the ARM and ST-ARM optimizer to its use in a sequential fashion, which was not originally considered. The lower performance of the $\ell_1$ regularized formulation is expected, as $\ell_1$ regularization is an approximation to the problem of selecting the optimal feature subset. In the following experiments, we have seen similar trends and only report the results from the Gumbel-Softmax based optimization.

Benchmarking results of different models are given in Table 2. As shown, our adaptive feature selection model is able to achieve a competitive accuracy using only 0.28% of the features, or on average about 1.57 sensors at any given time. We also observe that both the attention and our adaptive formulation is able to improve upon the accuracy of the standard GRU, suggesting that feature selection can also regularize the model to improve accuracy. Although the attention-based model yields the best accuracy, this comes at a cost of utilizing around 50% of the features at any given time.

We study the effect of the regularization weight $\lambda$ by varying it from $\lambda \in \{1, 0.1, 0.01, 0.005, 0.001\}$. We compare this with the attention model by varying the threshold $\alpha$ used to select features from $\alpha \in \{0.5, 0.9, 0.95, 0.99, 0.995, 0.999\}$, as well as the nonadaptive model by varying its $\lambda$ from $\lambda \in \{1000, 100, \ldots 0.01, 0.005, 0.001\}$. A trade-off curve between the number of selected features

Table 2: Comparison of various models for adaptive monitoring on three activity recognition datasets. *Accuracy metrics and average number of features selected are all in (%).

| Method | UCI HAR | | OPPORTUNITY | | ExtraSensory | | |
|---|---|---|---|---|---|---|---|
| | Accuracy | Features | Accuracy | Features | Accuracy | F1 | Features |
| Adaptive (Ours) $\lambda = 1$ | 97.18 | **0.28** | **84.26** | **15.88** | **91.14** | **55.06** | **11.25** |
| Attention $\alpha = 0.5$ | **98.38** | 49.94 | 83.42 | 54.20 | 90.37 | 53.29 | 54.73 |
| Nonadaptive $\lambda = 1$ [18] | 95.49 | 14.35 | 81.63 | 49.57 | 91.13 | 53.18 | 42.32 |
| No selection (GRU) [51] | 96.67 | 100 | 84.16 | 100 | **91.14** | 53.53 | 100 |

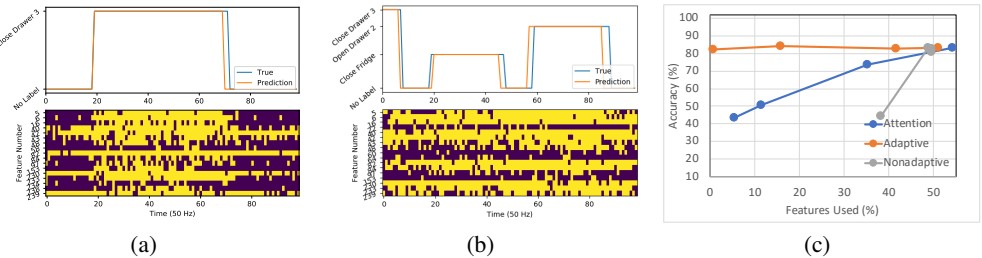

(a)                      (b)                      (c)

Figure 3: OPPORTUNITY Dataset results: (a) Prediction and features selected of the proposed model $\lambda = 1$. (b) Prediction and features selected of the proposed model on a set of activity transitions. (c) Feature selection vs. Error trade-off curve comparison.

and the performance for the three models can be seen in Figure 2(b). As shown in the figure, the accuracy of the attention model suffers increasingly with smaller feature subsets, as attention is not a formulation specifically tailored to find sparse solutions. On the other hand, the accuracy of our adaptive formulation is unaffected by the number of features, suggesting that selecting around 0.3% of the features on average may be optimal for the given problem. It further confirms that our adaptive formulation selects the most informative features given the context. The performance of the nonadaptive model is consistent for feature subsets of size 10% or greater. However, it suffers a drop in accuracy for extremely small feature subsets. This shows that for static selection, selecting a feature set that is too large would result in collecting many redundant features for certain contexts, while selecting a feature set that is too small would be insufficient for maintaining accuracy.

An example of dynamically selected features can be seen in Figure 2(a). We plot the prediction of our model compared to the true label and illustrate the features that are used for prediction. We also plot a heatmap for the features selected under each activity in Figure 2(c). Note that mainly 5 out of the 561 features are used for prediction at any given time. Observing the selected features, we see that for the static activities such as sitting, standing, and laying, only sensor feature 52 and 63, features relating to the gravity accelerometer, are necessary for prediction. On the other hand, the active states such as walking, walking up, and walking down requires 3 sensor features: sensor 65, 508, and 556, which are related to both the gravity accelerometer and the body accelerometer. This is intuitively appealing as, under the static contexts, the body accelerometer measurements would be relatively constant, and unnecessary for prediction. On the other hand, for the active contexts, the body accelerometer measurements are necessary to reason about how the subject is moving and accurately discriminate between the different active states. Meanwhile, we found that measurements relating to the gyroscope were unnecessary for prediction.

**UCI OPPORTUNITY Dataset**     We further test our proposed method on the UCI OPPORTUNITY Dataset [30]. This dataset consists of multiple different label types for human activity, ranging from locomotion, hand gestures, to object interactions. The dataset consists of 242 measurements from accelerometers and Inertial Measurement Units (IMUs) attached to the user, as well as accelerometers attached to different objects with which the user can interact.

We use the mid-level gesture activities as the target for our models to predict, which contain gestures related to specific objects, such as opening a door and drinking from a cup. A comparison of the accuracy and the percentage of selected features by different models is given in Table 2, while example predictions and a trade-off curve are constructed and shown in Figures 3(a), 3(b), and 3(c), with a similar trend as the results on the UCI HAR dataset. Notably, the trade-off for the nonadaptive models remains constant for $\lambda \in \{0.0001, 0.001, \ldots, 1\}$, with a sharp decrease in accuracy for $\lambda \geq 10$.

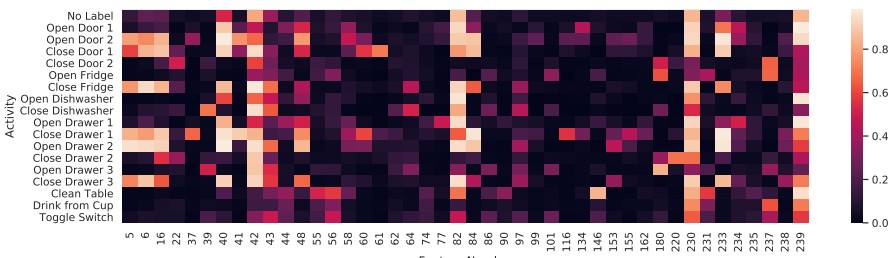

Figure 4: Heatmap of sensor feature activations under each activity of the OPPORTUNITY dataset. *Only active features are shown out of the 242 features in total.

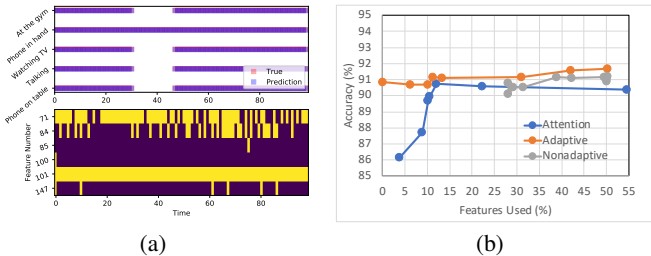

(a)                                    (b)

Figure 5: ExtraSensory Dataset results: (a) Prediction and features selected of the proposed model. (b) Feature selection vs. Error trade-off curve comparison.

A heatmap for the selected features under each activity is shown in Figure 4. Here, the active sensor features across all activities are features 40 and 42, readings of the IMU attached to the subject's back, feature 82, readings from the IMU attached to the left upper arm (LUA), and features 230 and 239, location tags that estimate the subject's position. We posit that these general sensor features are selected to track the subject's overall position and movements, as they are also predominantly selected in cases with no labels. Meanwhile, sensors 5, 6, and 16, readings from the accelerometer attached to the hip, LUA, and back, are specific to activities involving opening/closing doors or drawers.

Interestingly, sensors attached to specific objects, such as accelerometers on doors and cups, are unnecessary for prediction. We attribute this to the severe amount of missing values of these sensors. Indeed, the sensors that have the least amount of missing values are the body sensors and the localization tags. We hypothesize that the model prefers these sensors for their consistent discriminative power on multiple activity types compared to the object specific sensors. In addition to these object specific sensors, 5 IMUs, 9 accelerometers, and 2 localization tags can be completely turned off without significantly affecting prediction performance on this task.

**ExtraSensory Dataset**   We further test our proposed method on the ExtraSensory Dataset [31]. This is a multilabel classification dataset, where two or more labels can be active at any given time. It consists of 51 different context labels, and 225 sensor features. We frame the problem as a multilabel binary classification problem, where we have a binary output for each label indicating whether it is active. A comparison of the accuracy and selected features by different models tested can be seen in Table 2. Our method is again competitive with the standard GRU model using less than 12% of all the features.

A trade-off curve is shown in Figure 5(b), where we see a similar trend for both adaptive and attention models. However we were unable to obtain a feature selection percentage lower than 25% for the nonadaptive model even with $\lambda$ as large as $10^4$. We believe that this is because at least 25% of statically selected features are needed; otherwise the nonadaptive model will degrade in performance catastrophically, similar to the OPPORTUNITY dataset results. A heatmap and detailed discussion of the features that our model dynamically selected can be found in Appendix C.

The results on these three datasets along with the results on the NTU-RGB-D dataset in Appendix B indicate that our adaptive monitoring framework provides the best trade-off between feature efficiency and accuracy, while the features that it dynamically selects are also interpretable and associated with the actual activity types.

## 5   CONCLUSIONS

We propose a novel method for performing adaptive feature selection by *sequential context-dependent feature subset selection*, which is cast into a stochastic optimization formulation by modifying the $\ell_0$ regularized minimization formulation. To make this problem tractable, we perform a stochastic relaxation along with a differentiable reparamaterization, making the optimization amenable to gradient-based optimization with auto-differentiation. We apply this method to human activity recognition by implementing our method to Recurrent Neural Network-based architectures. We benchmark our model on four different activity recognition datasets and have compared it with various adaptive and static feature selection benchmarks. Our results show that our model maintains a desirable prediction performance using a fraction of the sensors or features. The features that our model selected were shown to be interpretable and associated with the activity types.

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

## A    IMPLEMENTATION DETAILS

Here, we provide the implementation details for the reported results in each benchmark dataset.

### A.1    UCI HAR DATASET

The UCI HAR dataset consists of a training set and a testing set. To implement our adaptive feature selection and other baseline methods, we divide the training set into a separate validation set consisting of 2 subjects. We preprocess the data by normalizing it with the mean and standard deviation. We then divide the instances of each subject into segments of length 200.

The base model we utilize is a one-layer GRU with 2800 neurons for the hidden state. We use the cross-entropy of the predicted vs. actual labels as the performance measure. We use a temperature of 0.05 for the Gumbel-Softmax relaxation. We optimize this with a batch size of 10 using the RMSProp optimizer, setting the learning rate to $10^{-4}$ and the smoothing constant to 0.99 for 3000 epochs. We then save both the latest model and the best model validated on the validation set.

### A.2    OPPORTUNITY DATASET

The OPPORTUNITY dataset consists of multiple demonstrations of different activity types. We first extract the instances into segments containing no missing labels for the mid-level gestures. Segments of length smaller than 100 are padded using the observed values at the next time-points in the instance. We then normalize the data such that its values are between -1 and 1. The authors of the dataset recommended removing some features that they believed are not useful, however we find that this does not affect performance and instead use the entire feature set. We have also experimented with interpolating the missing values but also find that it does not affect performance compared to imputing the missing values with zeros. Using this, we randomly shuffle the segments and assign 80% for training, 10% for validation, and 10% for testing.

The base model we utilize is a two-layer GRU with 256 neurons for each layer's hidden state. The cross-entropy of the predicted vs. actual labels is adopted as the performance measure. We use a temperature of 0.05 for the Gumbel-Softmax relaxation. We do not include the cross-entropy loss for the time points with missing labels. We also scale the total performance loss of the observed labels for each batch by $\frac{\#\text{timepoints}}{\#\text{labelled timepoints}}$. We optimize this loss with a batch size of 100 using the RMSProp optimizer, setting the learning rate to $10^{-4}$ and the smoothing constant to 0.99 for 3000 epochs. We then save both the latest model and the best model validated on the validation set.

### A.3    EXTRASENSORY DATASET

The ExtraSensory dataset consists of multiple demonstrations of human behavior under different activities, where two or more activity labels can be active at the same time. We first extract the instances into segments containing no missing labels for the middle level gestures. Segments of length smaller than 70 are padded using the observed values at the next time-points in the instance. We then normalize the data such that its values are in between -1 and 1. We have experimented with interpolating the missing values but also find that it does not affect performance compared to imputing the missing values with zeros. Using this, we randomly shuffle the segments and assign 70% for training, 10% for validation, and 20% for testing.

The base model we utilize is a one-layer GRU with 2240 neurons for its hidden state. We use a temperature of 0.05 for the Gumbel-Softmax relaxation. We use the binary cross-entropy of the predicted vs. actual labels as the performance measure, where the model outputs a binary decision for each label, representing whether each label is active or not. We do not include the performance loss for the missing labels and scale the total performance loss of the observed labels for each batch by $\frac{\#\text{timepoints} \times \#\text{total labels}}{\#\text{observed labels in labelled timepoints}}$. We optimize this scaled loss with a batch size of 100 using the RMSProp optimizer, setting the learning rate to $10^{-4}$ and the smoothing constant to 0.99 for 10000 epochs. We then save both the latest model and the best model validated on the validation set.

## A.4 NTU-RGB-D Dataset

We first preprocess the NTU-RGB-D dataset to remove all the samples with missing skeleton data. We then segment the time-series skeleton data across subjects into 66.5% training, 3.5% validation, and 30% testing sets. The baseline model that we have implemented for the NTU-RGB-D dataset is the Independent RNN [41]. This model consists of stacked RNN modules with several additional dropout, batch normalization, and fully connected layers in between. Our architecture closely follows the densely connected independent RNN of [41]. To incorporate feature selection using either our adaptive formulation or an attention-based formulation, we add an additional RNN to the beginning of this model. This RNN takes as input the 25 different joint features and is tasked to select the joints to use for prediction further along the architecture pipeline. Since the joints are in the form of 3D coordinates, our feature selection method is modified such that it selects either all 3 of the X, Y, and Z coordinates of a particular joint, or none at all. Our architecture can be seen in Figure 6.

Similar as the baseline method presented in [41], we have trained this architecture using a batch size of 128 and a sequence length of 20 using the Adam optimizer with a patience threshold of 100 iterations. We then save both the latest model and the best model validated on the validation set.

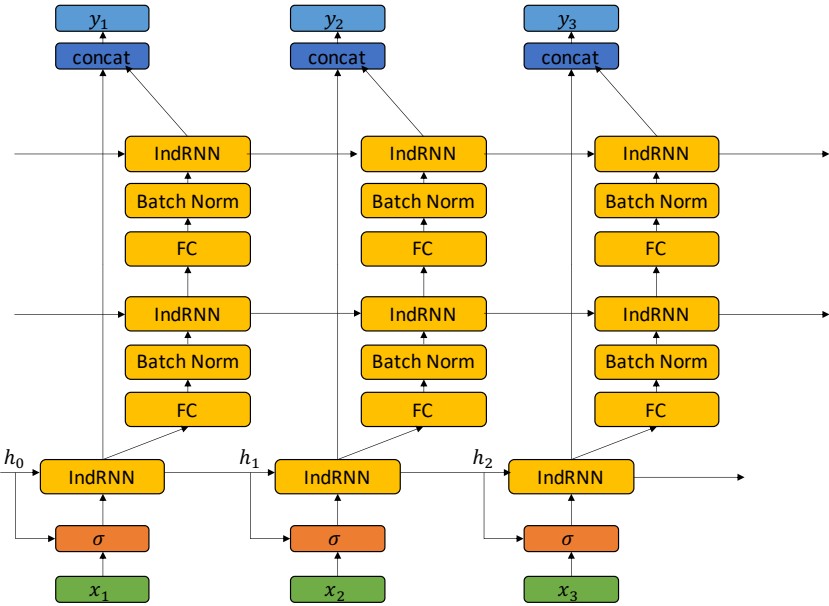

Figure 6: Our modified densely connected independent RNN architecture for adaptive feature selection.

## B Results and Discussion of the NTU-RGB-D Dataset

We have tested our proposed method on the NTU-RGB-D dataset [32]. This dataset consists of 60 different activities performed by either a single individual or two individuals. The measurements of this dataset are in the form of skeleton data consisting of 25 different 3D coordinates of the corresponding joints of the participating individuals.

We compare our method with three different baselines shown in Table 3: the standard independent RNN, a soft attention baseline, and a thresholded attention baseline. We see that our method maintains a competitive accuracy compared to the baseline using less than 50% of the features. On the other hand, because the thresholded attention formulation is not specifically optimized for feature sparsity, we see that it performs significantly worse compared to the other methods. Meanwhile, the soft-attention slightly improves upon the accuracy of the base architecture. However, as also indicated by our other experiments, soft-attention is not a dynamic feature selection method, and tends to select 100% of the features at all times.

A heatmap for the features selected under each activity is shown in Figure 7. Here, we can see that there are two distinct feature sets used for two different types of interactions: single person interactions and two person interactions. Indeed, since the two person activities require sensor measurements from two individuals, the dynamic feature selection would need to prioritize different features to observe their activities as opposed to single person activities.

Table 3: Comparison of various methods for activity recognition on the NTU-RGB-D dataset. *Accuracies and average number of features selected are in (%).

| Method | Accuracy | Features Selected |
|---|---|---|
| Adaptive | 80.54 | **49.65** |
| Thresholded attention | 40.07 | 52.31 |
| Soft attention | **83.28** | 100 |
| No selection | 83.02 | 100 |

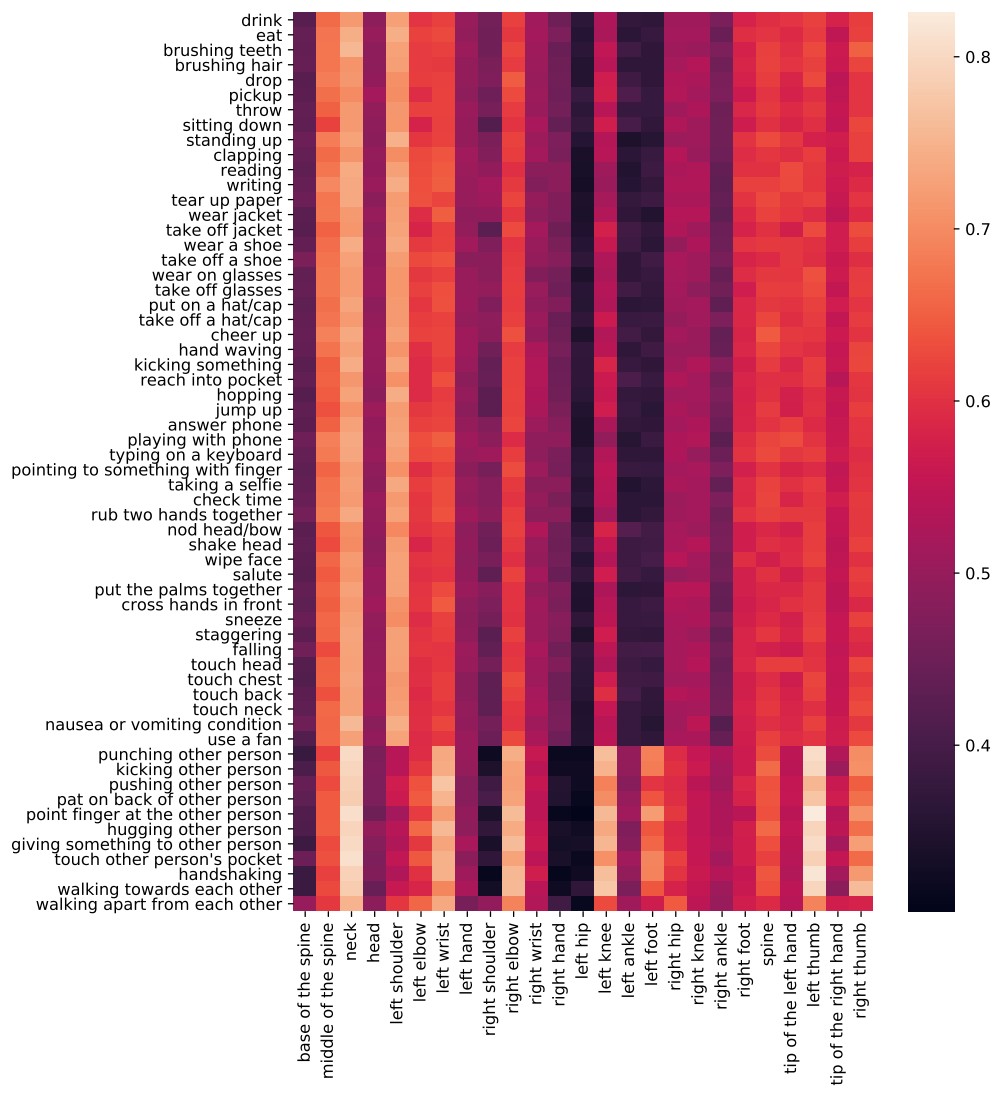

Figure 7: Heatmap of sensor feature activations under each activity state of the NTU-RGB-D dataset.

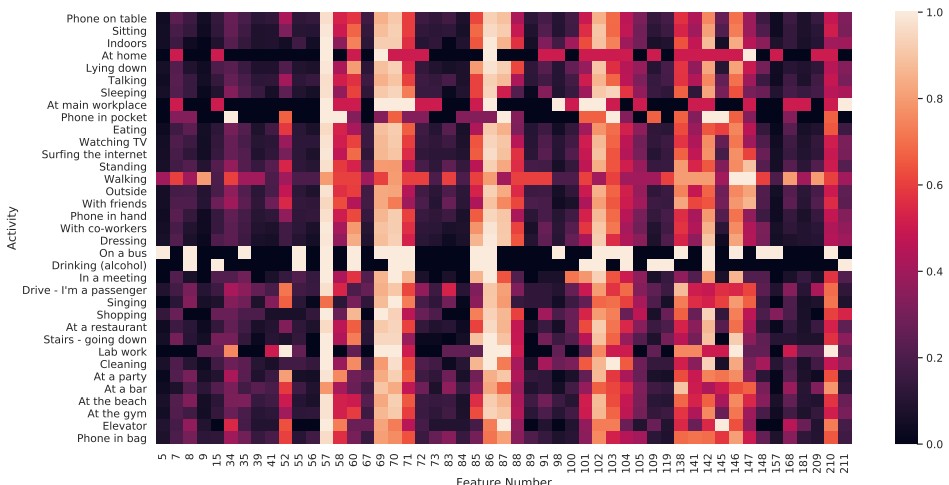

Figure 8: Heatmap of sensor feature activations under each activity state of the ExtraSensory dataset.

## C  RESULTS AND DISCUSSION OF THE EXTRASENSORY DATASET

A heatmap of the features selected under each activity state can be seen in Figure 8. As shown, there are four groups of sensor features that are used across activities: the phone magnetometer (57-71), watch accelerometer magnitude (85-88), watch accelerometer direction (101-105), and location (138-147). For two particular states, 'on a bus' and 'drinking alcohol', phone accelerometer measurements (5-52) become necessary for prediction. Some states such as 'at home', 'at main workplace', and 'phone in pocket' are notably sparse in sensor feature usage. We believe that these states are static, and do not require much sensor usage to monitor effectively. Other sensors such as the phone gyroscope, phone state, audio measurements and properties, compass, and various low-frequency sensors are largely unnecessary for prediction in this dataset.

