# OpenReview forum: "Dynamic Feature Selection for Efficient and Interpretable Human Activity Recognition"
_ICLR.cc/2021/Conference — Reject_

### Official Review · AnonReviewer4 · 2020-10-26
**I wonder if it can be applied directly to the online setting, which gradually decreases the number of features.**

**Rating:** 4
**Confidence:** 3

**Review:**

This paper proposes an RNN model for adaptive dynamic feature selection, for efficient and interpretable human activity recognition (HAR). From the intuition that human activity can be predictable by using a small number of sensors, the paper introduces an l0-norm minimization problem with parameter regularization, and provide a logic on formulating a dynamic feature selection model with relaxations. The difficulty of the discrete optimization problem is solved by differentiable relaxation, which is known as Gumbel-Softmax reparameterization techniques. The formulation is naturally led to an RNN model that uses histories as input with an additional sigmoid unit for adaptive feature selection.

Empirical studies are performed to show the superiority of the adaptive feature selection network. Results are shown on the task of 1) UCI-HAR smartphone dataset with 561 features, 2) UCI Opportunity sensor dataset with 242 features, 3) ExtraSensory dataset with 225 features for multilabel binary classification. In particular, by using the adaptive feature selection technique, the average number of features necessary for HAR prediction can be very small (0.3%, 15.9%, 11.3% among all features) at any given time. Overall, the paper is well written. In particular, analysis results on three datasets are clear and detailed, so that the reader would be available to understand what sensors were necessary for HAR prediction.

The key concern about the paper is that the algorithm lacks practicality. To show the adaptive selection algorithm is efficient, it should be shown that the algorithm drastically reduces features that are not necessary for prediction over time, while maintaining the performance even in the lighter feature space. Although the average number of features selected by the adaptive selection algorithm for each snapshot is small, all features are entered as input, which may not help to speed up the algorithm. To claim that the algorithm is efficient, it is required to show that the computation cost can be saved. Also, based on the current experimental results, it is difficult to say that features that were not used in earlier timestamp will not be used in later timestamp with a different context.

Minor comments and questions:
- Can you report the running time of each model?
- Is this model working in an online setting without tuning? If yes, would you like to clarify? If no, may I think this technique is for maintaining a dashboard that informs important features every time to users by calculating feature importance over time?
- The performance of the adaptive method on the NTU-RGB-D dataset is quite poor. What part of the dataset do you think caused the difficulty in feature selection? Do all features important?
- The technical novelty seems to be low if the proposed model is an RNN with an additional sigmoid layer.
- Figure 2a does not have a ground truth blue line.

---

> ### Author Response · Authors · 2020-11-14
> **Response to Reviewer 4**
>
> We thank the reviewer for the constructive comments and criticisms.
>
> Crucially, we clarify that our algorithm DOES NOT need all features to be entered as input. Using the previously observed feature sets, our model is able to select the next feature set to query without observing the entire feature set. As stated in our methodology section (equations (2), (3), (4), and (6)), the features that are not selected are set to zero and thus need not be observed. In practice, the next feature set to use can be determined without ever observing all of the features. We have shown through extensive experiments that our method drastically reduces the number of features that are not necessary for prediction for certain contexts.
>
> We also would like to point out the possible misunderstanding. Mostly importantly, our aim is not to have a compressed/simplified neural network model for activity recognition. Instead, we would like our dynamic feature selection framework to not only have good activity recognition performance, but also (more importantly) to inform which sensors should be turned on or off to achieve desired performance-cost tradeoff. Note that here the cost is incurred for sensor feature collection but not the inference for activity recognition based on given features.
>
> As we have stated in our manuscript, our claim on efficiency is not regarding computational efficiency, but rather efficiency in maintaining the sensors needed for prediction. Since not all sensors will be used at all times, we can use our gating method to safely TURN OFF sensors to achieve power efficiency.
>
>
> We answer the questions the reviewer listed below:
>
> - The running time of our model compared to a standard GRU is quite comparable. On the UCI HAR dataset, our model takes 11.15ms to predict each time point while a standard GRU takes 7.68ms. This is still within acceptable latencies for continuous activity monitoring.
>
> - The model learns a feature selection policy offline and uses the learned policy to select features during testing time. The parameters of the model and selection policy are fixed during testing and no online update of the parameters are done in this setting. The dynamic feature selection is achieved because the selection policy is conditioned on the previous observations to determine the next feature set to use. From this perspective, the technique can indeed be thought of as maintaining a dashboard indicating which features are important given the context summarized by the previous observations.
>
> - We note that we are still achieving competitive performance on the NTU-RGB-D dataset, with only about a 2.5% accuracy drop compared to the baseline model which uses the entire sensor set. We attribute this to the fact that the dataset only contains 25 sensors in total. We believe that such a small sensor set is already curated to be the most informative and therefore, any additional sensor selection would slightly degrade the performance of a predictive model.
>
> - We note that our model isn’t simply “an RNN with an additional sigmoid layer”. In fact, adding a simple sigmoid layer would not produce any feature sparsity by itself as shown by our comparisons with an attention-based model. Instead, in our model, the next feature set is sampled using a learned discrete probability distribution conditioned on the previous observations. Such a model isn’t trivial to optimize, as backpropagation isn’t trivially done through the discrete probability distribution parameters. Moreover, we also introduce a regularization that is easily evaluated using a stochastic relaxation. The resulting module is then highly applicable to more complex architectures and modeling scenarios.
>
> - In Figure 2a, the orange and blue lines both overlap exactly, causing the blue line to be unseen. We will fix this in our revision of our manuscript.
>
> We truly appreciate the reviewer to reply to our post and ask more questions to ensure there are no more misunderstandings that would have affected your assessment of our method. At this point, we are very confident that our above clarifications have thoroughly addressed the potential confusions and misunderstandings in the current review.

---

> > ### Comment · AnonReviewer4 · 2020-11-17
> > **Thanks for your answer, here is my follow-up thought.**
> >
> > Hello authors, thank you for clarifying my understanding. I have carefully read other reviews and your replies. While reading the manuscript, my main concern was the practical applicability of the solution you proposed. It is like the one that Reviewer 2 asked:
> >
> > "Given that this is what you measure since you still need all the features to use your model, what are the advantages of the method? Only interpretability?"
> >
> > -> First of all, I was reviewing the paper by considering applying this algorithm in an online setting (predefined model & online update). I understand the answers and your point that the features that are not selected are set to zero in this framework. As you responded, it will lead the device to turn off the sensors that relate to the features in practice. But once the sensors are turned off, there will be no more observations. Then, the training dataset for that feature will not be collected. This will lead you to this question: How can you let the model decide to turn it on again? There is no data for specific sensors.
> >
> > You were able to train a model since, in an offline environment that was considered, there was a dump training set that every feature values are recorded. Although the model (Eq. 3 and Eq. 6) can be trained by previous observations, can you guarantee that it will work in the long-term? Also, selecting training/validation/testing set by randomly shuffling 70%/10%/20% does not make sense. To build something that's robust, you may want to split data chronologically and check parameter stability over time.
> >
> > To conclude, I would like to thank the authors for clarifying some misunderstandings that I have, but I feel there is some gap between offline experiment settings and online settings. To guarantee that the proposed model is working in a real environment that requires HAR, it may require additional check-ups and justifications. Evaluating the algorithm with three datasets is a great starting point. However, I believe there is room to enhance the quality of the paper. The authors can consider advancing the current algorithm in 1) technically novel, or 2) practically applicable. For the former direction, I suggest you address comments from other reviewers. For the latter path, the applied science track of data mining conferences would also be suitable. They may want to see some results from user studies or A/B test results after deploying this solution to your products.

---

> > > ### Author Response · Authors · 2020-11-23
> > > **Response to Reviewer 4**
> > >
> > > We truly appreciate the reviewer’s timely reply to our responses.
> > >
> > > We first would like to clarify that the setting we consider in our paper is not an online learning/training setting where the model parameters are constantly updated. We instead learn the selection policy and prediction model offline and test it without parameter updates (neither the policy nor the prediction model). Nowhere in our paper did we state that we are considering the online learning setting. We would think our current setup is a fundamental and critical starting point, especially considering that there is no similar model or method exactly doing what we try to accomplish to the best of our knowledge, as well as after checking the suggested references based on all the reviewers’ comments. Further, we note that consideration of an online learning setting would not be very practical for human activity recognition systems, with or without considering dynamic feature selection. Even considering the case where we are able to observe all the features, it is unrealistic to assume that an end user would be able to label the activities in each instance reliably to help refine the model with better prediction performance. These would be interesting research directions that would need to consider non-stationarity of the signals, domain adaptation if any pre-trained model exists, as well as the extreme lack of labeled data.
> > >
> > > That being said, assuming that these complications are addressed, our model can be readily applied to the online setting, although this is beyond the scope of our paper due to the aforementioned complexity. To see this, we can think of our model as a reinforcement learning agent that is tasked to both give a prediction and select the next feature set to observe given the current belief state summarizing the previous observations. With this, we can see that optimizing (eq. 6) is equivalent to the reinforcement learning objective of minimizing the expected loss following a parameterized policy. While this is typically optimized through techniques such as policy gradient, we are instead estimating the gradients of the objective using the Gumbel-softmax relaxation. In general, we would recommend setting the temperature of the Gumbel-softmax to be lower than the one in the offline setting to enable feature sparsity even when not directly sampling discrete variables. Our model can then be trained by aggregating the data collected online, similar to [1]. We will consider investigating this possibility of online learning but we do believe that our presented work in the current submission is meaningful, practically applicable, and novel both in the problem we are considering and in the model formulation.
> > >
> > > Below, we clarify the additional misconceptions that the reviewer brought up:
> > >
> > > “How can you let the model decide to turn it on again? There is no data for specific sensors”. As we have stated, the model maintains a latent representation that enables previously turned-off features to be turned on again given the context. Our experiments clearly show that to be the case at test time, where some features are not selected to have the corresponding sensors turned on in the earlier time points but get selected to have the sensors turned on as the context changes. Note that the selection mechanism Eq. 7 only depends on the previously selected feature sets, and not all the features.  In the online learning setting, the model would start with a random feature selection policy. The randomness of our selection policy ensures that our model would explore using a variety of feature sets for a given context. This enables the model to gradually explore the state-action space before arriving at an optimal selection policy. Moreover, the randomness of our selection policy ensures that a well-optimized model would still occasionally explore using features it has previously deemed uninformative.
> > >
> > > “Although the model (Eq. 3 and Eq. 6) can be trained by previous observations, can you guarantee that it will work in the long-term?”. We appreciate the concern. As we explained, our current setup is assuming that the underlying dynamics are stationary and the training and testing data are from the same distribution, a common assumption in classical machine learning. Performance degradation can happen if there are internal or environmental changes. These are challenges that all machine learning models in a wide range of applications face when being deployed. As we stated at the beginning, these will be all interesting research problems involving transfer/self-supervised learning that we can explore within our presented framework. We refer the reviewer to [1] (Theorem 3.1) for guarantees in applying our algorithm to the online learning setting.

---

> > > > ### Author Response · Authors · 2020-11-23
> > > > **Response to Reviewer 4 continued**
> > > >
> > > > “Also, selecting the training/validation/testing set by randomly shuffling 70%/10%/20% does not make sense”. As we have stated, our random shuffles split data both chronologically and by different subjects. The human activity recognition (HAR) problem is a multi-class problem. To make sure that we evaluate HAR performance for all the activity types, we compared the performance with baselines in the current setup. To obtain the evaluation as suggested and cover all the activity types, it requires to check all the testing data and often depending on the datasets, it can be difficult to get such a comprehensive testing set. We also note that the UCI HAR dataset is divided into full trajectories of separate subjects in our experiments. The performance of our model on this dataset shows that indeed our model is stable for long-term predictions. To further illustrate this, we show the average accuracy over 1000 seconds of running the model on the testing subjects in this dataset. The results are shown below:
> > > >
> > > > Time: 0-999, 1000-1999, 2000-2999, 3000-3999
> > > >
> > > > Errors (%): 3.49, 2.93, 6.46, 4.06
> > > >
> > > > Std. Dev.: 1.89, 1.23, 1.05, 1.67
> > > >
> > > > Based on the results, there is no clear temporal degradation in the testing performance for this dataset. Instead, the change of prediction errors is mostly dependent on the underlying activity types.
> > > >
> > > > We thank the reviewer again for the reply, but we would like to firmly state our standing again that we believe our proposed work is a fundamental starting point and the first adaptive feature selection method for HAR to the best of our knowledge. Our method learns a feature selection policy offline, is practically applicable and achieves a comparable or better human activity recognition performance compared to existing baselines using a fraction of the sensor features on average.  We hope the above explanations have clarified the confusions and hope the reviewer could more positively evaluate our work accordingly.
> > > >
> > > > [1] Ross, Stéphane, Geoffrey Gordon, and Drew Bagnell. "A reduction of imitation learning and structured prediction to no-regret online learning." Proceedings of the fourteenth international conference on artificial intelligence and statistics. 2011.

---

### Official Review · AnonReviewer2 · 2020-10-27
**Review Dynamic Feature Selection**

**Rating:** 3
**Confidence:** 5

**Review:**

The authors tackle the important problem of feature selection. They propose to use differentiable gates with an RNN architecture to select different subsets of features for each time point. I think the idea and method are interesting, and the method could be useful. However, I have crucial problems with the way the paper is presented. Most importantly, the authors describe the l_0 relaxation of Bernoulli random variables as if it is their own contribution. They describe existing known results under a section titles “Methodology” as if they are the first to present Bernoulli random variables to feature selection or that they are the first to relax them using the Gumbel Softmax trick. They also use the word: “we derive” (p.3). This is wrong! And misleading! The same relaxation appears in [1] and used for model sparsification, the descriptions are almost identical to what appears in [1] with almost zero credit to the authors in [1] (a citation appears in related work in a different context). Bernoulli relaxation was already used for feature selection, in [2], and [3], these papers were not even mentioned. The reader can think the authors are the first to introduce such relaxation into the problem of feature selection, while this is again, clearly wrong.
The authors are well aware of that this relaxation was presented in [1], and in the experiment section they describe the baseline which solves (4) by citing [1] (citation [18] in their paper), this is again in contradiction to the way they describe the relaxation as if it is their own contribution.

Putting these CRITICAL comments aside, I think the results are misleading. Specifically, comparing the average number of selected features to the (constant) number of selected features of the non-adaptive method is misleading. You need to compare the union of selected features by your method to the constant number, otherwise, there is no way to infer if this feature selection method can result in any compression of the model or could lead to training or inference speed up.  Given that this is what you measure since you still need all the features to use your model, what are the advantages of the method? Only interpretability?

The authors do not explain how the method is used in the testing phase, is the randomness removed? How exactly?
The authors do not explain how training/ testing is performed, this appears in the appendix but should be moved to the main texts.
The authors should compare the method to the distribution suggested in [1], which seems more suitable for feature selection than the Concrete distribution (used by the authors).
Citations are not in the correct ICLR format.

Some pros: I like the examples used in the paper as well as the comparison to ARM, ST, ST-ARM.
To conclude, I am voting to reject the paper, based on all the reasons mentioned above.

[1] Louizos, Christos, Max Welling, and Diederik P. Kingma. "Learning Sparse Neural Networks through $ L_0 $ Regularization." ICLR, 2018.

[2] Yamada, Y., Lindenbaum, O., Negahban, S., & Kluger, Y.  Feature selection using stochastic gates. ICML, 2020.

[3] Balın, Muhammed Fatih, Abubakar Abid, and James Zou. "Concrete autoencoders: Differentiable feature selection and reconstruction." ICML. 2019.

---

> ### Author Response · Authors · 2020-11-14
> **Response to Reviewer 2**
>
> We thank the reviewer for the constructive comments and criticisms.
>
> We will clarify confusions and improve the presentation of our manuscript. Here, we would like to **strongly point out the possible misunderstanding by the reviewer**. Most importantly, our aim is **NOT** to have a compressed/simplified neural network model for activity recognition. Instead, we would like our dynamic feature selection framework to not only have good activity recognition performance, but also (more importantly) to inform which sensors should be turned on or off to achieve desired performance-cost tradeoff.
>
> Here the cost is incurred for sensor feature collection but not the inference for activity recognition based on given features. In mobile sensing tasks, devices are typically equipped with a redundant set of sensors, and turning on/off those sensors accounts for a major portion of on-device energy budgets. As Reviewer 1 has insightfully pointed out: “This has implications in energy consumptions of wearable sensors. but could even generalize to measurement timings in clinical care to make the work of nurses more efficient, and reduce the stress caused by some medical procedures.”
>
> We consider dynamic feature selection as  sequential context-dependent feature subset selection and cast it into a stochastic optimization formulation, enabling gradient-based solutions. The method we derived in our manuscript is a temporal feature selection method, which is different from the one presented in [1], where the goal is neural network architecture sparsification. Unlike in [1] where the weight masking is done independent of the current input data, our feature selection method is context dependent and adapts to the input data currently being handled. This introduces non-trivial complications when solving the optimization problem, especially when using the ARM optimization strategies. Also different from [1], our method selects features across time, while [1] is concerned with model compression. We note that many other methods in the literature can be derived as a relaxation similar to the one presented in [1] (see [a], [b], [c]). The fact that our method uses a similar relaxation as [1] should not discredit its novelty. It was never our intention to misrepresent our work and we will modify our write-up to more clearly show its relation to existing work.
>
> The other works [2] and [3] that the reviewer cited are also unlike the method in our manuscript. Crucially, both deal with static feature selection as opposed to the dynamic feature selection method we are proposing.
>
> We beg to differ from the reviewer’s statement that we should compare the union of the selected features to the constant number of features selected by the non-adaptive method. As we have stated numerous times in our manuscript, the problem we are concerned with is regarding DYNAMIC feature selection for continuous human activity recognition. Here, we are tasked to give a prediction for each time point while achieving sensor power efficiency by dynamically selecting the sensor set used for each time point. Because we are able to select fewer sensors on average, we are able to keep more sensors turned off on average thus achieving greater power efficiency. Comparing the union of the selected sensors would not be representative of the sensor power consumed by our method, as not all sensors in the union would be turned on at all times when using our method. Nevertheless, we have computed the union of features selected. All of them still show our method to be superior to the non adaptive method. We list these numbers below:
>
> UCI HAR: 3.56%.
>
> OPPORTUNITY: 19.83%.
>
> ExtraSensory: 26.66%.
>
> We emphasize that our model DOES NOT need all the features observed and to be entered as input. Using the previously observed features, our model is able to select the next feature set to query without observing the entire feature set. As stated in our methodology section (equations (2), (3), (4), and (6)), the features that are not selected are set to zero and thus need not be observed.
>
> In the testing phase, the randomness is not removed, we stated this in section 2.4. We would place additional details regarding training/testing in the main manuscript if the page limit permits. However, we believe that the training details would detract from our main message of achieving efficiency and interpretability using our model. We are currently revising both the main text and supplement to try our best to make these points clearer.
>
> We would truly appreciate the reviewer to reply to our post and ask more questions, if our points are still not clear enough, to ensure there are no misunderstandings regarding our method and motivations to develop dynamic/adaptive feature selection. At this point, we are very confident that our above clarifications have thoroughly addressed the potential confusions and misunderstandings in the current review.

---

> > ### Author Response · Authors · 2020-11-14
> > **References**
> >
> > [a] Wu, Bichen, et al. "Fbnet: Hardware-aware efficient convnet design via differentiable neural architecture search." Proceedings of the IEEE Conference on Computer Vision and Pattern Recognition. 2019.
> >
> > [b] V Campos, et al. Skip RNN: Learning to Skip State Updates in Recurrent Neural Networks. ICLR 2018.
> >
> > [c] van Baalen, Mart, et al. "Bayesian Bits: Unifying Quantization and Pruning." 2020.

---

> > ### Comment · AnonReviewer2 · 2020-11-22
> > **Severity of the "confusion" by author**
> >
> > I want to start by stressing the severity of what you term "confusion".
> > In your manuscript, you have presented existing results as if they are your own, you have not given proper credit to the creators of these results. This is in violation of a code of ethic by ICLR:
> > "Researchers should therefore credit the creators of ideas, inventions, work, and artefacts, and respect copyrights, patents, trade secrets, license agreements, and other methods of protecting authors' works."
> > I wasn't sure if I wanted to bring this up, but from your response, it seems that you are completely ignoring my main criticism. I believe that the way you falsely presented the $l_0$ relaxation as if it was your own could have biased the score given by reviewer 1.
> > Addressing the rest of your response:
> > If you claim that your method provides a computational advantage or power reduction why didn't you demonstrate such a result? The fact is that you've focused on providing performance plots vs. the number of selected features, these plots are MISLEADING. If your model is not using all features, you should indicate how many features it is using (by counting the union).
> > To summarize, given the fact that the authors did not address my main concerns about the false way they have presented the results of other authors, I can not raise my score.
> > I think the idea presented in the paper is worth a publication, but the authors need to drastically revise the way the method and results are presented.

---

> > > ### Author Response · Authors · 2020-11-23
> > > **Disagree with your serious accusation of “violation of a code of ethic” (Please avoid emotional and ungrounded criticism)**
> > >
> > > While we respect the discussion on problem formulation we are disappointed and, frankly, shocked at such an accusation. To reiterate from our prior response (paragraph 3), we claim that our problem and formulation are new regarding dynamically selecting features based on the underlying states/contexts, and we tried different ways to explain this in our responses in addition to our efforts in our original submission.
> > >
> > > We never claimed that the $l_0$ relaxation solution is our own and we have cited all the papers that deal with stochastic optimization when involving discrete decision variables to our knowledge, including the original straight-through idea, and the relaxation methods including Gumbel-softmax, as well as more recent methods including ARM.
> > >
> > > Note that we have attributed the static l0 formulation to Louizos et al. (2017) (or [18]) in the Results table (Table 2) and in the following location in the original submission file: “To show the effect of considering dynamic feature selection, we compare a nonadaptive l0 formulation that statically selects features by solving (4) [18]”.  Although [18]’s goal is not feature selection, due to the similarity of the optimization strategy, we have attributed the static feature selection by relaxation of l0 and the solution to Eq. (4) to that work. Clearly, we are not trying to get credit for the l0 relaxation under static setup with global parameters.
> > >
> > > Further, in Section 2.4 of our original submission, we clearly referred to the Gumbel-Softmax (Concrete) papers. In Sections 3 and 4, we have extensively discussed the different methods for optimization involving discrete variables in neural networks and cited relevant papers. We would like to note that we are currently revising our paper based on all four reviewers’ comments. If there are sentences in our submission that made the reviewers or other readers feel that we invented these relaxation techniques, please kindly point out these places so that we are aware of these. We feel insulted by getting such an accusation.
> > >
> > > In the original submission file, in Section 2.3, the sentence starting with “To extend ...” is our contribution regarding relaxation of the adaptive feature selection formulation. In the revised manuscript that we upload now (but still trying to revise to make sure that we address all the concerns from all four reviewers), we have added the following sentences below Eq (5) to prevent any potential confusion for readers unfamiliar with the literature that the reviewer is concerned about:
> > >
> > > “Relaxation of binary random variables has been adopted in Louizos et al. (2017) for network architecture
> > > sparsification, and in Yamada et al. (2019); Balın et al. (2019) for static feature selection. Here, we
> > > extend the above relaxation for time series data, where unlike previous works, the binary random
> > > variables are parameterized locally and are context-dependent, and features are selected adaptively
> > > across time.”
> > >
> > > We are sorry if it is not intuitive that the number of features will be positively correlated with the number of required sensors, and therefore directly reflect the power usage. Reporting the average number of features used has been done by other previous works in similar setups (see [d, e] or [2, 42] in the paper). We did check the union of the selected features for three datasets as mentioned in our reply above, which still show our method to be superior to the non-adaptive method. Please do refer to our first reply “Response to Reviewer 2”. We also post the statistics here for easier reference:
> > >
> > > UCI HAR: 3.56%.
> > >
> > > OPPORTUNITY: 19.83%.
> > >
> > > ExtraSensory: 26.66%.
> > >
> > > An example of including what the reviewer had asked, i.e. providing the union of all features, in our direct response to the comments, and the reviewer again asking for the same result, leads us to the conclusion that the reviewer has not carefully or completely read our paper and responses. We felt that the reviewer misread and tried to mis-present our paper based on his/her own reading, which is not only careless but also unethical itself to impose this baseless accusation.
> > >
> > > We are indeed trying to revise our submission based on all four reviewers’ comments at this moment. We surely welcome all your suggestions on how we shall present our method and results but would appreciate your constructive suggestions instead of accusing us without giving the actual examples where we violated the code of ethics.  Since we feel that the reviewer now brought the discussion to quite an emotional state and put ungrounded yet overly harsh criticism to our work (we’re not sure why, though), we politely urge the area chair to take this factor into account and step in this discussion, if possible.
> > >
> > > [d] Yang, Xiaodong, et al. "Instance-Wise Dynamic Sensor Selection for Human Activity Recognition." (2020).
> > >
> > > [e] Strubell, Emma, et al. "Learning dynamic feature selection for fast sequential prediction." (2015).

---

> > > > ### Comment · AnonReviewer2 · 2020-11-23
> > > > **Reply to authors**
> > > >
> > > > I have carefully read the paper, numerous times. I am extremely familiar with the field and with the existing results. I have reviewed numerous papers in this field, and have never seen authors mention work developed by others under a section titled "Methodology", without explicitly mentioning that what they are presenting is existing work (which should be under background).
> > > >
> > > >
> > > >
> > > > "We felt that the reviewer misread and tried to mis-present our paper based on his/her own reading, which is not only careless but also unethical itself to impose this baseless accusation."
> > > > In my initial review, I kindly requested the authors to address this concern, however, this was not fully addressed in their response.
> > > >
> > > > In the original version these are the locations the authors gave credit for existing results presented under the methodology section (citing the authors):
> > > > "Note that we have attributed the static l0 formulation to Louizos et al. (2017) (or [18]) in the Results table (Table 2) and in the following location in the original submission file: “To show the effect of considering dynamic feature selection, we compare a nonadaptive l0 formulation that statically selects features by solving (4) [18]”. Although [18]’s goal is not feature selection, due to the similarity of the optimization strategy, we have attributed the static feature selection by relaxation of l0 and the solution to Eq. (4) to that work. "
> > > > -Credit is only given in the results section, so if a reader only reads the Methodology section, the relaxation is attributed to the authors. And Credit was not given to Yamada et al. which use the relaxation for feature selection.
> > > >
> > > > I thank the authors for addressing my concern in the current version they due give credit at the appropriate location.
> > > >
> > > >
> > > > "Since we feel that the reviewer now brought the discussion to quite an emotional state and put ungrounded yet overly harsh criticism to our work (we’re not sure why, though), we politely urge the area chair to take this factor into account and step in this discussion, if possible."
> > > > I apologize if this has offended the authors, I do not claim that this was intentional, however, the presentation of the methodology was misleading, especially for readers that are not familiar with this line of works.
> > > >
> > > > Thanks for pointing out the union of features, I think this is substantial and should be emphasized in your revised versions.

---

> > > > > ### Author Response · Authors · 2020-11-24
> > > > > **Reply to Reviewer 2**
> > > > >
> > > > > We thank the reviewer for spending significant time on reading our submission and responses multiple times. We also apologize if we have offended the reviewer somehow either in our original submission or our first response. We are glad to come to a mutual understanding.
> > > > >
> > > > > We would, however, now draw the reviewer’s attention to our latest draft. Frankly, we suspect our submission was not assessed on its merits due to the previous concerns raised by the reviewer. In particular, we draw attention back to three points:
> > > > >
> > > > > 1) The reviewer originally did not seem to get what we tried to claim that our dynamic/adaptive feature selection formulation is new and different from the existing work. We got this impression as the reviewer compared our method to the reference [1] in regards to model compression/inference speed up in his/her original review. We tried to explain the possible confusion in our first response because of this impression.
> > > > >
> > > > > 2) Regarding the $l_0$ relaxation accusation, we again clearly stated that we “extend” the existing non-adaptive/static solutions to help solve our adaptive formulation in Section 2.3 of the original submission. We would like to again note that it is critical to present Eq. 6 in methodology to derive adaptive and locally parametrized feature selection, which is essential for our paper. We presented the non-adaptive static formulations there for better readability to make sure that the readers can get the differences between Eq. 4 and Eq. 6. As we have shown in our ongoing revision, we took the reviewer’s original review seriously and have tried to address this concern by adding more detailed discussion as well as references in Section 2.3. We did not post the revision with our first response as we were waiting for all the additional experimental results to address all the reviewers’ comments. We welcome more suggestions if the reviewer felt that our current revision still has issues or statements that somehow annoy the reviewer.
> > > > >
> > > > > 3) The reviewer misunderstood our purpose of dynamic feature selection in the original review stating “there is no way to infer if this feature selection method can result in any compression of the model or could lead to training or inference speed up.” We tried to explain this in our first response that we have never tried to compress or simplify the activity recognition model in the first place. Our purpose is to reduce the sensor and thereafter power usage, which was presented in our original submission.
> > > > >
> > > > > Again, we are open to any constructive suggestions on how we shall further improve our presentation and truly appreciate any concrete and constructive requested changes, including reorganizing the specific content, that we may need to revise further. Meanwhile, from the current discussion, it has been very clear that:
> > > > >
> > > > > - Our current paper has no major technical flaw, and several misunderstandings were already clarified.
> > > > > - We have not over-claimed our contribution. The relationship between our submission and related work has been addressed, from the first submission draft, to the current discussion thread.
> > > > > - We’re open to more paper structure discussions and appreciate your inputs, but those only re-organized what we already have had in paper.
> > > > >
> > > > > We politely request the reviewer to:
> > > > >
> > > > > - Take back his/her ungrounded accusation on “violating code of ethics”. That is too serious for us to accept.
> > > > > Consider providing another fair re-assessment of this work based on the above clarification discussions.
> > > > >
> > > > > - Let’s all lower the temperature and focus back on the science - we very respectfully thank the reviewer’s time and would sincerely appreciate a more positive consideration.

---

### Official Review · AnonReviewer3 · 2020-10-29
**Although this paper is well written and reported results are positive, the novelty of this paper is quite limited. Besides, several highly-related previous works are missing, and important hyper parameter studies are not reported. These problems prevent me from rating the paper as acceptable.**

**Rating:** 4
**Confidence:** 4

**Review:**

This paper presents a learning-based binary sampling mechanism for feature selection. It filters salient feature dimensions by sampling from a Gumbel-softmax distribution, which is differentiable and can be trained with other network parameters. The proposed method is evaluated on several Human Activity Recognition (HAR) datasets.

The positive and negative points of this paper can be summarized as following:

pros:
+ This paper is well written and is easy to follow.
+ The experimental evaluations give positive results.

cons:
- Important previous works are missing. Learning to generate categorical samples for RNNs is not a fresh idea. Actually, [a] has already employs Gumbel-softmax to sample scales in order to dynamically control the temporal pattern learning; More generally, the topic of this paper is connected to a amount of previous works aiming to adaptively decide how/when to memorize/update the inputs/states, such as [b] and [c]. These works should also been cited by this paper.

- With these missing works taking into account, the novelty of this paper becomes incremental and contribution is trivial. Integrating Gumbel-softmax sampling with RNN cells is very straightforward, and the motivation of applying Gumbel-softmax is very similar to [a]. While [a] is proposed for general sequence tasks, the proposed method seems to work only for HAR with multi-dimensional inputs.

- Since \tau is the only hyper parameter of Gumbel-softmax, evaluations on how the value of \tau could impact the performance can be important. Yet no such results are reported in the paper. From the original Gumbel-softmax paper we can see a sample can approximate to a one-hot vector when \tau is small and be closed to a uniform distribution when \tau goes large. So it is very likely that the performance will become unstable as \tau changes. Showing such experimental results could be definitely improve the paper quality. I would suggest to report the means and stds of accuracies with different sampling seeds.

Summary:
Considering the concerns listed above, I believe there are problems that outweighs the strengths of this paper. They should be fixed before acceptance.

[a] H Hu, et al. Learning to Adaptively Scale Recurrent Neural Networks. AAAI 2019
[b] V Campos, et al. Skip RNN: Learning to Skip State Updates in Recurrent Neural Networks. ICLR 2018
[c] D Neil, et al. Phased LSTM: Accelerating Recurrent Network Training for Long or Event-based Sequences. NIPS 2016

---

> ### Author Response · Authors · 2020-11-14
> **Response  to Reviewer 3**
>
> We thank the reviewer for the constructive comments and criticisms.
>
> We emphasize that our method is a dynamic sensor selection method that infers which **small subset of sensor features** to use **adaptively**, at any given context or state indicated by the latent representation, **at any given time point**. In this way, its aim is to optimize the tradeoff between prediction accuracy and power and/or other incurring cost to maintain the sensor features,  It is unlike any of the methods the reviewer has cited.
>
> [a] selects wavelet scales used for prediction, instead of which features to use at any given context. This is motivated by a need to capture multi-scale patterns to achieve a more expressive model, instead of for energy efficiency or interpretability.
>
> [b] selects which time steps to skip for updating the recurrent state, and applies this to sequence modeling tasks. This may not be appropriate for continuous monitoring tasks, where we are tasked to give a prediction at every time step. Moreover, it doesn’t enable interpretability of the model’s decisions. Indeed, with our model, we can maintain a small adaptive set of sensors for easily discriminable contexts, and we can observe these sensors for interpretation of the model’s behavior.
>
> [c] introduces a gating mechanism across time which controls the frequency with which the memory cell is updated. Such a selection is not context based, as the parameters of the gating mechanism, such as the period and the phase shift, are independent of the current state of information given in order to optimize both prediction accuracy and sensor usage. On the other hand, our dynamic selection mechanism is context dependent, it uses the previous observations to infer the next feature set to use in order to optimize both the prediction accuracy and sensor usage.
>
> We further note that all the references applied selection or skipping along the temporal direction, which is only loosely related to our work, to say the least. Our proposed dynamic feature selection focuses on which sensor features across time points should be selected to achieve the desired performance-cost tradeoff and the feature selection is adaptive with respect to the underlying states or contexts of the corresponding activities and environment. Our dynamic/adaptive feature selection is orthogonal to temporal selection/skipping and we do believe these are two different research questions. But we will be happy to cite those papers as our broader context, and we can explore the potential of integrating these two directions as our future research.
>
> Gumbel-softmax is one strategy to solve our proposed optimization formulation (the true originality and merit): we neither claimed it as our originality, nor consider it the only way to go; in face, we also state that the Gumbel-softmax along with many other categorical selection mechanisms have been used for many purposes such as attention, model compression, mixture of experts, Neural Turing Machines, etc. Usage of such a selection mechanism alone should not be grounds to say that a method is not novel. We would consider these are potential solution strategies to achieve our proposed dynamic/adaptive feature selection. Indeed, we benchmarked multiple methods to optimize our dynamic feature selection formulation and found that the Gumbel-softmax performed most favorably for our applications.
>
> We have tested different hyperparameters \tau ranging from 0.001 to 5 and did not notice any significant differences in performance. We will shortly update our manuscript to include our comparisons on \tau in the appendix.
>
> We hope the above can clarify several confusions and make our novelty & merit much more clear to the reviewer.

---

### Official Review · AnonReviewer1 · 2020-11-03
**Very good paper, but some tuning required in its claims.**

**Rating:** 9
**Confidence:** 5

**Review:**

The authors provide a novel combination of known architectures to an important use case of reducing the density of required  measurements in sensor-fusion based temporal multi-class inference tasks. This has implications in energy consumptions of wearable sensors.  but could even generalise to measurement timings in clinical care to make the work of nurses more efficient, and reduce the stress caused by some medical procedures..

The authors represent a way to train consistent policy that predicts the best combination of sensors to estimate the state of the subjects.  They have found that a smaller set of features.  is more explainable than the full set of features.  However, I think that this somewhat of an overpromise.  The trained model gives the optimal density of the measurements and can discern also if old values and features measured are till OK for the inference.  This does not mean that those measurements are not needed at all in the features.  One can only argue that the required features can be estimated from the older measurement. So, the current set of active sensors is not the full set of required measurement values and can not be exclusively used to explain the logic of the system. Even more, the logic of the policy deciding the new  measurement is not discussed in an explainability context. The authors provide no data on this. It may be just an  estimate the derivative of the signal and ignore a new measurement, if it's time  derivative is small enough.

As a summary , I support publication of the manuscript, provided  the authors modify the message on the interpretable features.

---

> ### Author Response · Authors · 2020-11-23
> **Response to Reviewer 1**
>
> We thank the reviewer for the constructive comments and criticisms.
>
> We agree that in the current formulation, “one can only argue that the required features can be estimated from the older measurement. So, the current set of active sensors is not the full set of required measurement values and can not be exclusively used to explain the logic of the system.” Indeed, some features can potentially be reliably estimated from older measurements, and our method can decide to turn off these sensors depending on the underlying states or contexts and save energy. Using the previously observed features, our model is able to select the next feature set to query without observing the entire feature set. We tried to explain this by showing that the selected features indeed reflect the actual activities. As the reviewer pointed out, our dynamic/adaptive feature selection is learned using all the training data. Showing the activity class specific features can be considered as a retrospective interpretation of what our method can do. For testing, dynamic feature selection is determined by the derived latent representations based on all the past observations. We will revise to better explain this point.

---

### Decision · Program_Chairs · 2021-01-07
**Final Decision**

**Decision:**

Reject

**Comment:**

The authors propose a methodoloy for dynamic feature selection. They use differentiable gates with
an RNN architecture to select different subsets of features at each time point thus resulting in dynamic selection.
The reviewers agree that the idea is interesting and the method could be useful and I share their opinion.

The majority vote is towards rejection. The overarching mwssage of the reviews is that the manuscript raises confusion in a number of points. I see this work as one with good potential for impact but its current presentation is confusing. The vivid discussion that it raised is also an indication of it. The authors have done a good job replying to the concerns and the questions raised. However, the reviewers were still unsatisfied with the authors response to their concerns.  I recommend rejection at this time, while encouraging the authors to take seriously the reviewers' requests for a clearer presentation of their approach's contribution in order to strengthen their paper for future submission.